

# Closure analysis of aerosol-cloud composition in tropical maritime warm convection

Ewan Crosbie[1,2], Luke D. Ziemba[2], Michael A. Shook[2], Claire E. Robinson[1,2,3], Edward L. Winstead[1,2], K. Lee Thornhill[1,2], Rachel A. Braun[4,†], Alexander B. MacDonald[4,‡], Connor Stahl[4], Armin Sorooshian[4,5], Susan C. van den Heever[6], Joshua P. DiGangi[2], Glenn S. Diskin[2], Sarah Woods[7], Paola Bañaga[8], Matthew D. Brown[1,2], Francesca Gallo[2,9], Miguel Ricardo A. Hilario[5], Carolyn E. Jordan[2,10], Gabrielle R. Leung[6], Richard H. Moore[2], Kevin J. Sanchez[2], Taylor J. Shingler[2], Elizabeth B. Wiggins[2]

[1]Science Systems and Applications, Inc., Hampton, VA 23666, USA.
[2]NASA Langley Research Center, Hampton, VA 23666, USA.
[3]Department of Chemistry, Collage of William and Mary, Williamsburg, VA 23187, USA.
[4]Department of Chemical and Environmental Engineering, University of Arizona, Tucson, AZ 85721, USA.
[5]Department of Hydrology and Atmospheric Sciences, University of Arizona, Tucson, AZ 85721, USA.
[6]Department of Atmospheric Science, Colorado State University, Fort Collins, CO 80523, USA.
[7]Stratton Park Engineering Company (SPEC), Boulder, CO 80301, USA.
[8]Manila Observatory, Quezon City, 1108 Philippines.
[9]NASA Postdoctoral Program, Oak Ridge Associated Universities, Oak Ridge, TN 37831, USA.
[10]National Institute of Aerospace, Hampton, VA 23666, USA.

† Now at: Healthy Urban Environments Initiative, Global Institute of Sustainability and Innovation, Arizona State University, Tempe, AZ 85287, USA.
‡ Now at: Department of Environmental Sciences, University of California, Riverside, CA 92521, USA.

*Correspondence to*: Ewan Crosbie (ewan.c.crosbie@nasa.gov)

**Abstract.**

Cloud droplet composition is a key observable quantity that can aid understanding of how aerosols and clouds interact. As part of the Clouds, Aerosols and Monsoon Processes – Philippines Experiment (CAMP²Ex), three case studies were analyzed involving collocated airborne sampling of relevant clear and cloudy airmasses associated with maritime warm convection. Two of the cases represented a polluted marine background, with signatures of transported East Asian regional pollution, aged over water for several days, while the third case comprised a major smoke transport event from Kalimantan fires.

Sea salt was a dominant component of cloud droplet composition, in spite of fine particulate enhancement from regional anthropogenic sources. Furthermore, the proportion of sea salt was enhanced relative to sulfate in rainwater and may indicate both a propensity for sea salt to aid warm rain production and an increased collection efficiency of large sea salt particles by rain in subsaturated environments. Amongst cases, as precipitation became more significant, so too did the variability in the sea salt to (non-sea salt) sulfate ratio. Across cases, nitrate and ammonium were fractionally greater in cloud water than fine-





mode aerosol particles; however, a strong co-variability in cloud water nitrate and sea salt was suggestive of prior uptake of nitrate on large salt particles.

A mass-based closure analysis of non-sea salt sulfate compared the cloud water air-equivalent mass concentration to the concentration of aerosol particles serving as cloud condensation nuclei for droplet activation. While sulfate found in cloud

was generally constrained by the sub-cloud aerosol concentration, there was significant intra-cloud variability that was attributed to entrainment – causing evaporation of sulfate containing droplets – and losses due to precipitation. In addition, precipitation tended to promote mesoscale variability in the sub-cloud aerosol through a combination of removal, convective downdrafts, and through dynamically-driven convergence. Physical mechanisms exerted such strong control over the cloud water compositional budget that it was not possible to isolate any signature of chemical production/loss using in-cloud

observations. The cloud-free environment surrounding the non-precipitating, smoke case indicated sulfate enhancement compared to convective mixing quantified by a stable gas tracer; however, this was not observed in the cloud water (either through use of ratios or the mass closure) perhaps implying that the warm convective cloud time scale was too short for chemical production to be a leading-order budgetary term and because precursors had already been predominantly exhausted. Closure of other species was truncated by incomplete characterization of coarse aerosol (e.g., it was found that only 10-50%

of sea salt mass found in cloud was captured during clear air sampling) and unmeasured gas phase abundances affecting closure of semi-volatile aerosol species (e.g., ammonium, nitrate and organic aerosols) and soluble volatile organic compound contributions to total organic carbon in cloud water.

## 1 Introduction

Clouds play an important global role in the production, loss and redistribution of atmospheric aerosol particles, as well as

altering their physical, chemical and optical properties. Precipitation is the dominant removal process and therefore exerts a strong governing influence on the lifetime of aerosol particles (Textor et al, 2006; Wang et al., 2020). Incorporation of aerosol mass into cloud water through activation of cloud condensation nuclei (CCN) – known as "nucleation scavenging" – (Jensen and Charlson, 1984) and the contribution from impaction/diffusional uptake of interstitial particles by cloud droplets (Flossman et al., 1985), leads to subsequent removal, subject to physical conversion of the cloud condensate to surface precipitation –

known as "rainout" (Radke et al., 1980; Flossman et al., 1985). Alternatively, cloud and rain drops can release the scavenged material upon evaporation (Mitra et al., 1992; Wang et al., 2020) or collect additional material in subsaturated environments (e.g., below cloud base); however, this mechanism – known as "washout" – has reportedly variable importance (Bae et al., 2012; Aikawa and Hiraki, 2009; Andronache, 2003; Croft et al., 2009) depending on both aerosol particle size and rain rate (Andronache, 2003).


Cloud droplets are recognized as important reaction sites for the production of low volatility products such as sulfate (Hegg and Hobbs, 1982; Lelieveld and Heintzenberg, 1992; Hegg and Larson, 1990) and secondary organic aerosol (SOA), formed through aqueous reactions (Blando and Turpin, 2000; Ervens et al., 2011; McNeill, 2015; Lim et al., 2010). Additionally, cloud processing alters the chemical properties of aerosols and degree of oxidation (Chakraborty et al., 2016; Lee et al., 2011;

Ervens et al., 2018) and exerts microphysical changes that can affect size (Hoppel et al., 1994; Feingold et al., 1996) and mixing state/size-composition relationships (Riemer et al., 2019; Hegg et al., 1992; O'Dowd et al., 1999). Combined, these





processes have downstream impacts on bulk aerosol characteristics – such as hygroscopicity (Jimenez et al., 2009; Shingler et al., 2016; Sorooshian et al., 2017) and optical properties (Hegg et al., 2004; Eck et al., 2012) – that can influence aerosol interactions with subsequent clouds (Feingold and Kreidenweis, 2000). Gases of varying solubility may dissolve into cloud
droplets, undergo subsequent chemical alteration and then return to the gas phase upon cloud drop evaporation (e.g., Laj et al., 1997; Marinoni et al., 2011) and gas-particle partitioning may be affected by the cloud droplet passage through cloud (Hayden et al., 2008). Solute composition affects cloud pH, and thereby subsequent acid deposition by precipitation (Shah et al., 2020), but also exerts a controlling influence on aqueous chemistry (Scott, 1978; Collett et al., 1994; Kreidenweis et al., 2003; Hegg and Larson, 1990; Ervens et al., 2011) and the partitioning of semi-volatile species (Pye et al., 2020).


In addition to production and loss mechanisms, clouds act as a conduit for vertical redistribution of particles (e.g., Corr et al., 2016; Wonaschütz et al., 2012; Baumgardner et al., 2005; Reid et al., 2019; Leung and van den Heever, 2022) and trace gases (e.g., Dickerson et al., 1987; Fried et al., 2016; Li et al., 2018; Bela et al., 2016; Yin et al., 2001; Mari et al., 2000). In tropical maritime settings, convective clouds are critical to the energy and moisture budget and vertical transport is tightly coupled to
diabatic processes associated with radiative and latent heating (Riehl and Malkus, 1958; Johnson et al., 1999; Sobel and Bretherton, 2000). This challenges attempts to directly observe aerosol removal processes through sampling of cloud-free air, since convergent inflows into regions of enhanced cloudiness have yet to directly "feel" cloud effects, while the analysis of low-level divergent outflows (i.e., associated with unsaturated downdrafts and subsequent cold pools) is confounded by difficulty distinguishing downward transport of background air aloft from specific rainout processes.


Mixing between cloudy and surrounding clear air parcels is ubiquitous (Romps and Kuang, 2010; de Rooy et al., 2013), and is inherent to the dispersive role that clouds play. Entrained aerosols and gases may contribute to cloud droplet composition, while at the same time, the incorporation of unsaturated air into cloud may affect the scavenged fraction. For example, a homogeneous mixing process that distributes evaporative effects across the droplet population (Jensen and Baker, 1989)
enhances solute concentrations and retains scavenged mass, while inhomogeneous mixing (resulting in complete evaporation of a subset of droplets) would tend to lower the scavenged fraction, all else equal. The effects of dilution by the surrounding environment presents an added source of variability (e.g., beyond achieving closure of terms affecting absolute abundances inside and outside cloud), since relative composition has been found to vary with droplet size (e.g., Bator and Collett, 1997) and the nature of mixing (e.g., homogeneous/inhomogeneous) is strongly size-dependent (Burnet and Brenguier, 2007).


Except under "natural laboratory" experiments, such as ground-based orographic studies (e.g., Fowler et al., 1988; van Pinxteren et al., 2016) and mountain waves (Hegg and Hobbs, 1982), it is difficult to separate cloud-processed airmasses from the unperturbed environment. In convective clouds, parcel trajectories are highly variable and estimating cloud contact time is non-trivial, despite efforts to establish "cloud clock" markers (Witte et al., 2014). This complex interplay amongst
production, loss and redistribution is succinctly highlighted in Koch et al. (2003), who find anticorrelation (at daily to monthly



timescales) between cloudiness and observed surface sulfate across Europe and North America, indicative of stronger removal of sulfate by precipitation and suppression of gas-phase production than enhancement of aqueous production pathways. In a modelling study, Berg et al. (2015) showed the important contribution of the collective cloud effects on aerosol at the sub-grid scale (in their case ~10km) relating to both shallow and deep convection, through up to 50% reductions of black carbon (a

primary aerosol tracer) – ostensibly attributed to enhanced redistribution and rainout in precipitating convection – and up to 40% enhancement in sulfate associated with shallow cumulus. Spatial and temporal heterogeneity of clouds and precipitation, and the resulting impact on aerosols, confounds efforts to understand how aerosols exert control over cloud microphysical properties and precipitation (e.g., Gryspeerdt et al., 2015) , as does the co-variability of aerosols and the environment (e.g., Varble et al. 2018), and the modulation of aerosol impacts on microphysical processes by environmental characteristics such

as CAPE, moisture and instability (Storer et al. 2010; Khain et al. 2005; Lee et al. 2008; Grant and van den Heever 2015; Fan et al. 2009; Marinescu et al. 2021).

Despite the complexity, and potential circularity, of using limited airborne and ground-based observations to untangle the dynamical, microphysical and chemical processes governing cloud composition, it is clear that further systematic observations

are needed comprising collocated aerosol, gas and cloud composition, ideally in concert with idealized cloud-chemistry box models and cloud-resolving simulations. Recent studies based on airborne field experiments incorporating direct measurements of marine cloud droplet chemical composition have focused on regional surveys (e.g., Benedict et al., 2012; Straub et al., 2007; Sorooshian et al., 2018); the abundance and pathways of specific cloud species such as: nitrate (Prabhakar et al., 2014), amines (Youn et al., 2015), organosulfur compounds (Sorooshian et al., 2015), carboxylic acids (Hegg et al.,

2002; Crahan et al., 2004; Sorooshian et al., 2013), and trace elements (Wang et al., 2014; Mardi et al., 2019); vertical structure (MacDonald et al., 2018); coupling with the ocean surface (Wang et al., 2016); relationships with cloud droplet number concentration (MacDonald et al., 2020); and, dynamical features (Crosbie et al., 2016). The Clouds, Aerosols and Monsoon Processes – Philippines Experiment (CAMP[2]Ex) offered a rare opportunity to conduct airborne sampling in tropical maritime convective environments with an extensive suite of aerosol, cloud and radiation measurements in combination with modelling

studies. Here we present three case studies of tropical maritime warm convection that cumulate airborne sampling of aerosol composition immediate below and surrounding the cloud systems with cloud and rain drop composition at multiple levels. We report boundary layer aerosol composition and examine properties in the context of local variability associated with the cloud systems, and the imprint they impart on aerosol properties through scavenging, mixing, and transport. We then discuss closure of non-sea salt sulfate mass between the aerosol and cloud, selected for its ubiquity and relatively constrained budget, in the

context of physical interactions, in-cloud production, and instrument sampling limitations. We follow this with a discussion of the cloud composition reporting the drivers of variability (within and amongst cases) and the relationship with the aerosol composition.



## 2 Methods

### 2.1 Clouds, Aerosols and Monsoon Processes – Philippines Experiment (CAMP$^2$Ex)

A total of nineteen research flights were conducted aboard the NASA P-3 aircraft based at Clark International Airport, Philippines during August-October 2019, targeting a diverse range of aerosol and cloud environments throughout the region. A focus area of the mission was the complex interaction amongst aerosol properties (composition, optical, and microphysical), monsoon clouds, and radiation; with flights designed to sample clear and cloudy airmasses using a combined payload of remote sensing and in situ instrumentation.   Here we consider three case studies that relate to flights that took place on September

19-20, 2019 (Case I), September 23-24, 2019 (Case II) and September 15-16, 2019 (Case III), selected for their combination of extended low-level in situ sampling – used for characterization of aerosol and trace gases in the sub-cloud mixed layer – with systematic sampling of warm (liquid only) maritime convective clouds at multiple altitudes.

### 2.2 Aerosol Measurements

The following listing is not exhaustive and only includes the instruments used in this analysis. A particle-into-liquid sampler

(PILS; Brechtel Manufacturing Inc.) continuously sampled ambient aerosol producing a liquid stream that was first passed through a conductivity cell (henceforth, PILS-Conductivity; Crosbie et al., 2020) followed by fractional collection, for offline analysis (Sorooshian et al., 2006).  The PILS setup did not include upstream denuders nor an impactor on the inlet; specifically done to aid the detectability of sea salt and other large particles (e.g., dust). The PILS-Conductivity, described in full in Crosbie et al. (2020), can be used as an independent cross-comparison to offline ion analysis and provides a continuous proxy measure

for total solute not afforded by batch sampling.  Non-refractory aerosol mass concentrations (< 1 μm) were measured using a high-resolution time-of-flight Aerosol Mass Spectrometer (AMS; Aerodyne Research Inc.) and refractory black carbon (BC) mass concentrations were measured using a Single Particle Soot Photometer (SP2; Droplet Measurement Technologies).  Dried (RH < 40%) and humidified (RH ≈ 80%) total scattering (450, 550 and 700 nm) were measured using two integrating nephelometers (TSI Model 3563).  Particle size distributions were measured using an Aerodynamic Particle Sizer (APS; TSI

Model 3321, 500-5000 nm diameter) and a Laser Aerosol Spectrometer (LAS; TSI Model 3340, 100-3000 nm diameter) that was size-corrected assuming a particle refractive index of ammonium sulfate (1.52+0i; Moore et al., 2021). The LAS sample flow was actively dried using a Nafion dehumidifier, while the APS was passively dried and located close to the sampling manifold to maximize transmission of super-micrometer particles.  Integrated volume and number in sub-micrometer (LAS) and super-micrometer (LAS and APS) size ranges were used as proxies for comparison with compositional mass-based

measurements.  Condensation particle counters (CPC; TSI Model 3756 and 3772, respectively) provided measurements of ultrafine (CN$_{>3 nm}$) and fine (CN$_{>10 nm}$) total particle concentrations, and an additional CPC (TSI Model 3772) downstream of a thermal denuder at 350°C provided non-volatile particle concentration (CN$_{>10 nm, nv}$).





During flight, ambient aerosols were continuously drawn through an isokinetic inlet (McNaughton et al., 2007) connected to
a manifold that supplied sample flow to all instruments. Data from the isokinetic inlet reported here were screened for periods
free from cloud and precipitation in order to avoid artifacts from shatter and resuspension. During cloud penetrations, the flow
delivered to a subset of instruments – that included the AMS, LAS and SP2 – was manually switched to sample from a
counterflow virtual impactor inlet (CVI; Brechtel Manufacturing Inc; Shingler et al. 2012) to characterize properties of cloud
residual particles.  Data from the CVI sampling periods were not investigated as part of this study, but times when these
instruments were diverted to the CVI result in gaps in the ambient dataset, even during times when cloud penetrations were
not occurring.

## 2.3 Cloud Measurements

While flying through cloud, discrete samples of cloud water (CW) were collected using an axial cyclone cloud water collector
(AC3; Crosbie et al., 2018).  The AC3 continuously separated cloud and rainwater from the airstream and diverted the collected
liquid into a sample line, subsequently pumped into storage vials for offline laboratory analysis. The system includes metering
of the sample collection rate allowing tracing of the CW sample to its environment.  Each sample can only be interpreted as
representing the bulk cloud environment (i.e., aggregated across the sample duration) and performance analysis has shown
reduced collection efficiency for smaller droplets (D < 20 µm; Crosbie et al., 2018), thus cases spanning this size range may
be expected to exhibit some bias towards larger droplet sizes.  Historically, airborne collection of CW has been dominated by
low-altitude sampling of stratiform boundary layer clouds (Sorooshian et. al., 2018), where conditions are usually relatively
homogeneous and the envelope of cloud microphysical properties quite limited. Due to the number of degrees of freedom
(droplet size distribution, transit time, liquid water content (LWC), temperature, altitude/pressure, flight speed and airframe
mounting details) the performance envelope (e.g., size-dependent collection efficiency, evaporative losses) remains theoretical
for regions of the parameter space sparsely populated by flight data (see Crosbie et al. 2018 for details).  CAMP²Ex has added
significantly to the dynamic range of microphysical conditions and environments for CW collection. In summary, AC3 was
found to be generally ineffective for collecting cloud water samples in polluted shallow cumulus that were characterized by
very small droplet sizes, low LWC, and short transit times (< 5 s), but very effective for collecting samples in (i) developing
cumulus turrets with high LWC even if the crossing was short (~10 s), (ii) clouds containing precipitation, and (iii) unsaturated
rain shafts (e.g., those traversed below cloud).  The AC3 was shuttered while clear of cloud (or precipitation) to minimize
impaction of coarse-mode particles that may be washed off and collected during a subsequent cloud penetration.  Prior to each
flight, the AC3 exposed interior surfaces were rinsed thoroughly with ultrapure water and several (2-4) blank samples were
collected pre- and post-flight by misting the collector with ultrapure water to simulate cloud water collection and assess
collector and laboratory artifacts.

Cloud microphysical measurements were made using a suite of wing-mounted probes and we include here the specific probes
that were used to estimate LWC and generate a merged drop size distribution across the full size spectrum.  In the cases



described here, all were sufficiently below the melting level that liquid drops were universally assumed. The Fast Cloud Droplet Probe (FCDP; SPEC Inc.) provided droplet size spectra from 1.5-50 μm, 2D-S Probes (SPEC Inc.; Lawson et al., 2006), configured in the 10 μm and 50 μm versions, provided stereo optical array images that were used to determine droplet

spectra from 5-3000 μm and > 25 μm, respectively, and the High-Volume Precipitation Spectrometer (HVPS; SPEC Inc.) was used to provide spectra for sizes greater than 150 μm. A duplicate set of measurements that together comprised a "Hawkeye" package, provided redundancy for the FCDP and 2D-S (10μm) datasets, but the Hawkeye FCDP was only used during instances of missing data. Drop size distributions were merged onto a gridded distribution with logarithmic bins and each contributing instrument was prescribed weights spanning its individual bin size range with tapered tails to smooth the transition

between contributing instruments. A graphical summary of the stitching weights is provided in the Supplement (Figure S1). Spherical volume was assumed to derive volume distributions and a water density of 1 g cm⁻³ was used during integration to estimate a time series of LWC. Estimates of cloud droplet number concentration were taken as the total number concentration measured by the FCDP. The FCDP was also used for characterization of coarse aerosol during sampling of clear air (LWC < 0.001 g sm⁻³, no precipitation, RH < 95%). This provided particle number and volume estimates (i.e., analogous to those

described above for the LAS and APS) at ambient conditions extending to larger sizes to aid characterization of coarse aerosol.

In addition, integration of the volume distribution above a threshold (D > 100 μm) was used to quantify a time series of rain water content (RWC), and a non-dimensional rain water fraction (RWF = RWC / LWC) designed to isolate regions of cloud where sedimentation processes were significant, and CW samples may disconnect from their airmass properties. Precipitation

rate (P) was estimated using size-dependent drop terminal velocity data (Beard 1976) integrated over the drop volume size distribution. Since this represents a higher-order moment of the size distribution, there is a tendency to amplify noise caused by low counting statistics for large rain drops, despite use of the HVPS at large diameters offering a significant benefit. P was used for support in classifying each case through the following computations: (i) a cloud-mean rain rate estimate, encompassing all time in cloud (LWC > 0.1 g sm⁻³) or rain (RWC > 0.001 g sm⁻³), (ii) a cloud base rain rate estimate as (i) but only considering

data collected near or below cloud base, and (iii) a "peak" rain rate estimated from the mean of the top quartile.

## 2.4 Auxiliary Airborne Datasets

Trace gas concentrations were used to aid the identification of airmasses and to provide support for assessing processes affecting the aerosol and cloud composition and their respective budgets. CO, CO₂ and CH₄ were measured using a near-IR cavity ringdown spectrometer (DiGangi et al., 2021) and O₃ was measured by a dual-beam ultraviolet absorption sensor (2B

Technologies, Model 205), all at ~2 s interval. Water vapor was measured using an open path Diode Laser Hygrometer (Diskin et al., 2002) and temperature was obtained from measurements of total air temperature using a Rosemount 102 probe. Three-dimensional wind components were derived using a radome-mounted, inertially-corrected 5-hole gust probe (Thornhill et al., 2003; Barrick et al., 1996).





## 2.5 Laboratory Analysis

A field laboratory was set up at the Clark International Airport to conduct post-flight chemical analysis of the PILS and CW samples. Chemical analyses were completed during the course of the field campaign usually within three days of the flight, with samples refrigerated before analysis. Both sample sets were analyzed using ion chromatography (IC) for selected inorganic and organic anions and cations. Two complete anion and cation IC systems were deployed to the field to manage the quantity of analysis (IC1: ICS-3000, ThermoFisher Scientific, IC2: ICS-2100, ThermoFisher Scientific). The systems used

AS11 and CS12 columns for anions and cations, respectively. For convenience, because of autosampler compatibility with the fraction collector vials, IC1 was dedicated to the PILS samples, which were exclusively analyzed with IC, while IC2 analyzed aliquots of the CW samples. Program run times for IC1 were shorter than IC2 to accommodate the significantly higher sample count which, combined with typically lower aqueous ion concentrations, resulted in fewer species reported for PILS ($Na^+$, $NH_4^+$, dimethylaminium (DMA), $K^+$, $Mg^{2+}$, $Ca^{2+}$, $Cl^-$, $NO_2^-$, $Br^-$, $NO_3^-$, $SO_4^{2-}$, oxalate) than CW (additional species: glycolate,

acetate, formate, methanesulfonate (MSA), pyruvate, glutarate, adipate, succinate, maleate). A set of freshly prepared ion standards were run periodically to ensure stability throughout the field campaign and to maintain consistency between IC1 and IC2.

    Aliquots of CW samples with sufficient remaining sample volume (total sample volume > 2 mL) were analyzed for pH

(Thermo Scientific Orion 8103BNUWP ROSS Ultra). The meter was calibrated with pH 4.0 and 7.0 buffer solutions before each batch of samples (typically encompassing 1-2 flights). Following the pH aliquot, and volume permitting (total sample volume > 5 mL), remaining CW was then analyzed for total organic carbon (TOC). TOC analysis (Sievers 800 Turbo TOC Analyzer) was conducted in larger batches, three times during the campaign. The TOC analyzer was calibrated using oxalic acid solutions before and after the sample batch and zeroes (ultrapure water) were taken between each sample. All CW was

collected in 15 mL centrifuge tubes (Corning) and the tubes were triple-rinsed with ultrapure water, soaked for at least 24 hours in ultrapure water and then triple-rinsed another three times during pre-flight (1-2 hours before takeoff).

## 2.6 Airborne Sampling Strategy

    CAMP[2]Ex implemented multiple flight strategies to meet the requirements of a combined in situ and remote sensing payload. Objectives involving clouds and cloud penetrations often required maneuvering and track adjustments based on the evolving

environmental conditions. The collection of CW was therefore linked to the amount of time spent conducting in situ cloud sampling and the properties of the sampled clouds (e.g., LWC, horizontal extent, and cloud microphysics). CW samples were manually advanced once sufficient volume was collected, or if a period of time (typically a few minutes) and distance vertically or horizontally had elapsed, such that collecting a new sample was desirable. In some instances, multiple CW samples were collected in a continuous block within the same contiguous cloudy region, while in other cases a single CW sample comprised

partial volumes from several discrete cloud penetrations. Sample volumes varied from <1 mL to 15 mL.



## 2.7 Cloud water sample merge

Laboratory analyses on CW samples provided information about the bulk aqueous properties, that reflected a weighted integration over the duration of the sample. For budget and closure analyses, aqueous concentrations need to be converted to an air-equivalent mass (AEM) using a characteristic LWC, and it is desirable to merge auxiliary airmass properties (e.g., trace gases) and dynamic conditions (e.g., statistics of 3-D wind fluctuations) onto each sample. Figure 1 shows an illustrative example of the environment surrounding typical CW collection during CAMP$^2$Ex, and highlights the rapidly changing airmass properties associated with a transect through an active convective turret near its top. Within this convective element, there were at least three local maxima that all contributed to the sample, the final being the most active core with the highest LWC, strongest updraft, and highest enhancement of boundary layer tracers (CO, CH$_4$). The sample metering system indicates where the CW was collected (Figure 1d), broadly tracking LWC, although some of the features were comingled. A threshold (LWC > 0.1 g sm$^{-3}$) was used to integrate (average) candidate properties, selected based on findings from previous field campaigns (Crosbie et al. 2018), and sampling in small cumulus during CAMP$^2$Ex, that 0.1 g sm$^{-3}$ represents an approximate lower bound for CW collection; however, it is recognized that the collection efficiency is size dependent and therefore may influence the threshold. Merged properties (LWC, RWC, N$_d$, vertical velocity (w), trace gases) were calculated and reported for each CW sample (Table S1) and a threshold sensitivity test (i.e., by using a range of LWC thresholds and a further merge using a weight proportional to LWC) was performed to determine uncertainty in the merge (Table S2). To first order, the rate of accumulation of CW scales with LWC (i.e., ignoring size/collection efficiency effects) thus a LWC-weighted average is more akin to integrating over the sample volume than integrating over time; however, we have no prior knowledge of how solute concentration varies with respect to LWC (e.g., at cloud edges) therefore it is likely that such an averaging scheme would overestimate the AEM when cloud core conditions represented a more dilute (e.g., more water for the same solute) environment. Covariance amongst the merged quantities at time scales shorter than CW samples represents an uncertainty for closure analysis (until instrument improvements allow the solute variability across these timescales to be diagnosed or directly measured). From Table S2 we can conclude that, while many CW samples are relatively insensitive to the merge/threshold method, there could be up to a factor of 2 uncertainty in deriving AEM, with high uncertainties tending to coincide with instances when a sample spanned a wide range of conditions such as a convective element embedded within a stratiform layer, or mixtures of cloud and rain within a single sample.

## 2.8 Compositional groups

Sea salt (SS) mass was calculated as the sum of the mass concentration of Na$^+$, Cl$^-$, Mg$^{2+}$ plus fractional contributions of K$^+$, Ca$^{2+}$, and SO$_4^{2-}$ derived from the sea water mass ratio of each species to Na$^+$ (0.037, 0.04, 0.25, assigned respectively) unless such contribution exceeded the total measured, in which case the total was used. This approach is grounded in the assumption that Na$^+$ is an exclusive tracer for SS. Non-sea salt (nss) contributions were then assigned to the remaining mass of K$^+$, Ca$^{2+}$, and SO$_4^{2-}$ ($_{nss}$K$^+$, $_{nss}$Ca$^{2+}$, $_{nss}$SO$_4^{2-}$). These groups are used to describe both PILS and CW data. AMS species/groups are



expressed without the charge (e.g., AMS NH$_4$) for readability so as to distinguish from the direct measurement of ions by PILS or CW.

**2.9 Airmass trajectories**

Back trajectories were generated using the Hybrid Single Particle Lagrangian Integrated Trajectory Model (HYSPLIT; Stein et al., 2015) at 1 min temporal resolution along relevant flight tracks, then averaged across designated airmasses. Input meteorological data were from the Global Forecast System reanalysis at a horizontal resolution of 0.25° × 0.25°. Transport analysis on flight data was performed using the method described in Hilario et al. (2021).

**3 Case Descriptions**

Case I: September 19-20, 2019

During this flight, intensifying Tropical Storm Tapah was situated approximately 600 km northeast of the northern coast of Luzon (23.02°N, 127.18°E; IBTrACS). A swath of cloudiness associated with a broad band of warm convection was located
north of Luzon along an axis approximately WNW-ESE, aligned with the inflow low-level circulation of Tapah, and served as the regional focus of the flight. A second line of convection extended to the southwest approximately perpendicular to the main line and through the course of the flight, the satellite presentation of this second line became progressively more disorganized. Evidence of episodic cold pool outflow boundaries, marked by arcs of shallow cumulus, could be observed in visible satellite imagery through the course of the flight. An overlay of the flight track atop a visible satellite image taken at
0330 UTC September 20, 2019 from the Advanced Himawari Imager (Figure 2a) illustrates the sampling strategy in connection with the main convective line and approximately corresponds to the temporal mid-point of the cloud sampling. The primary convective mass associated with Tapah is just off image to the NE and the disorganized, northern extent of the second convective line is seen interacting with the western extent of the main study area.

The section of the flight track highlighted red indicates the cloud module (CM) which encompassed a cloud "wall" pattern comprising multiple, sequentially-descending, level legs at selected cloud altitudes along the NW-SE axis of the convective line, immediately followed by sampling below the cloud. The sub-cloud legs were arranged approximately perpendicular (SW-NE) to the cloud legs to assess any cross-track gradients and to increase time outside precipitation for aerosol sampling. The locations of CW collected within the CM are indicated by the cyan circles, illustrating the proximity and high-density of
the samples. Approximately 150 km to the SE, low altitude (<2 km) sampling was performed earlier in the flight (highlighted green) in an environment that contained both fair weather cumulus and more vertically developed shallow convection, but did not have the same areal extent of stratiform/detrained cloud cover from 1.5-3 km that characterized the environment of the CM. The sub-cloud sampling conducted during this period (the downwind survey) provided a direct comparison with the





perpendicular transect on the upwind flank of the CM. Spiral profiles extending above cloud top (highlighted blue) were
conducted at the beginning and end of the downwind survey, and upon the completion of the CM. Horizontal wind vectors
(20 s mean) are shown (Figure 2a) on the crosswind sub-cloud legs for both downwind (green) and upwind (red), noting that
a marked increase in wind speed was observed in the downwind region, presumed to be attributed to the greater influence of
Tapah. A cloud top wind vector (cyan) was calculated using the average of the first (highest) CM leg.

In the vicinity of the CM, the lifting condensation level (LCL) – used to estimate the location of the lowest cloud bases – was
estimated at 680 m and a representative cloud top height was 3.5 km (Table 1). Given that the scene was evolving, the cloud
top height estimate is merely a snapshot broadly capturing conditions at the time of the CM; there were nearby cells beyond
the sampling area with tops extending above 4 km. Most of the local cloud top maxima appeared (visually) as undulations in
the extensive stratiform cloud, which was sampled during two of the CM legs and resulted in the majority of the CW samples.
At lower altitudes, cloud coverage was more broken and clustered around cumulus feeding the upper stratiform layers.
Precipitation was widespread and was encountered within active cells, between clouds and below cloud base with a peak rain
rate of 5.8 mm hr$^{-1}$ and mean cloud base rain rate 4.0 mm hr$^{-1}$.

Case II: September 23-24, 2019


The same color scheme and features (as described above) were used to annotate the flight track overlaid on an equivalent
satellite image (0400 UTC Sep 24, 2019, see Figure 2b). The primary cloud system studied during this case was smaller in
spatial extent than Case I, but similarly organized as a linear feature with an axis close to that of the low-level wind (NE). In
contrast to Case I, vertical shear of the horizontal wind was observed as a directional shift in cloud top winds to northwesterly
(above approximately 3 km). The study feature was rooted within a broader area of enhanced shallow convection, while
surrounding regions mainly comprised patches of cloud-free and fair-weather cumulus fields. There was no observed maritime
deep convection within approximately 500 km.

As per Case I, this flight involved a "wall" pattern CM; however, cloud tops were somewhat higher (~4 km) and more vertical
levels were sampled (7). Upon completion, there was an extended survey within the mixed layer offset from the CM to
coordinate sampling with the nearby ship, the R/V Sally Ride, and this pattern provided ample time to fully characterize nearby
regions and assess mixed layer spatial variability. A spiral profile followed in largely cloud-free conditions (scattered shallow
cumulus) to complement the data collected in clear adjacent regions during the CM (this was more widespread compared to
Case I, because of the compactness of the cloudy region). The estimated cloud liquid water path (LWP) was highest in this
case as was the peak LWC (5.6 g m$^{-3}$), observed near cloud top.

Case III: September 15-16, 2019



This flight was dedicated to sampling and characterization of transported biomass burning emissions from Kalamantan fires.

A low-altitude survey of the smoke plume was performed approximately perpendicular to the wind (i.e., from SE to NW) to capture source variability through measurements of trace gas and aerosol abundances. Subsequently the aircraft repositioned approximately 300 km downwind to probe vertically-developed cumulus. Here the CM comprised a spiral descent in proximity to one of these cells with penetrations through the cloud at seven levels (six of which yielded CW samples) and, near cloud base, the aircraft then offset to a cloud-free region for a second vertical profile. AHI visible satellite imagery (with flight track

overlay and markups, per previous description; Figure 2c) shows the cloud scene at the time of the descending spiral (0400 UTC, Sep 16, 2019). There were far fewer CW samples collected during this case as a result of significantly less time spent in cloud, but each sample corresponded to a single cloud transect. An additional CW sample (yellow marker) was collected near the top of a developing cumulus cloud (4.3 km) shortly after completion of the smoke transect and is included for additional context. The cloud system studied in Case III was distinct from Cases I and II in that (i) convection was isolated, not organized;

and (ii) precipitation was negligible in the lower half of the cloud. Despite the cloud top being the highest of the three cases (estimate 4.8 km), the development of precipitation appeared to be inhibited by entrainment, which also curtailed high LWC maxima. Both Case II and III had similar environmental RH and convective turbulence (as quantified by extrema in vertical winds), yet the apparent impact of the environment on Case III was more pronounced, likely because of its smaller size and earlier stage of maturation. Case III was associated with a significantly higher aerosol abundance as a result of the smoke, and

that was observed to affect the cloud microphysics (highest $N_d$ of the three cases), which may suppress warm rain processes (Feingold et al. 2013; Tao et al., 2012) and also enhance entrainment (e.g., Jiang et al. 2006). Low-level winds, from the smoke survey upwind, indicated confluent SW flow across the Sulu Sea and winds near cloud top indicated low shear across the target cloud system (Figure 2c); however, there were clear indications from satellite imagery and the aircraft forward-facing video camera that other more vertically-developed cells existed in the vicinity of the CM and those were affected by

northerly shear.

## 4. Results and Discussion

4.1 Boundary layer aerosol

Sub-cloud sampling during Case I included the downwind survey prior to the CM and the upwind transect that formed the final leg of the CM. The upwind transect afforded a simple divide into respective airmasses north (UN) and south (US) of precipitation associated with the main cloud line, while the downwind survey identified northern (DN) and southern (DS)

airmasses based on distinct changes in properties, yielding four airmass quadrants (Figure 2a) relevant to the analysis of the cloud complex. Evaluation of the time-series during the downwind survey (Figure 3) showed that below cloud (< 600 m) there




were three periods (highlighted, grey background) where aerosol enhancements were coincident with reductions in CO and
CH₄. Review of the spatial patterns of these gradients, and the associated horizontal wind anomalies (for further details see
Figure S2), provided support for the marked distinction of quadrants DN (aerosol enhanced) and DS (CO enhanced) bestriding
a region of confluent flow. The first two crossings into DN occurred during the spiral descent providing snapshots at two
altitudes and exhibited a warmer, drier and well-mixed sub-cloud layer, viewed as a regional background environment
compared to the cooler, moister and stratified environment of DS, potentially indicative of influence by recent precipitating
convection. The jump in CO and CH₄ across the boundary was sharp, while the aerosol gradient was diffuse. Comparisons
with concurrent timeseries of O₃ and CO₂ (Figure S2) show that O₃ exhibited a similar diffuse gradient and its abundance
correlated positively with aerosol, and with O₃-poor conditions in DS, while CO₂ changed sharply across the boundary
analogous to CO and CH₄. Elevated CO₂ in DS showed significant fine-scale variability (and the highest peak concentrations
observed across the sampling region) indicative of local sources, perhaps attributable to active fumarole emissions from the
nearby Babuyan Islands. Mean properties of the airmass quadrants are summarized in Table 2.

The AMS composition indicated that the non-refractory aerosol mass was predominantly SO₄, with OA:SO₄ ratio varying
between 0.18-0.28. The highest organic contribution occurred in DS coincident with the enhanced CO, BC and non-volatile
number fraction (i.e., CN>10nm,nv /CN>10nm) which may indicate that this airmass was more influenced by primary combustion
aerosols perhaps originating from transported sources in Luzon or locally, from the Babuyan Islands; however, all airmass
quadrants exhibited high $f_{44}$ (the ratio of m/z 44 to total OA signal, which is a marker for the organic aerosol degree of
oxygenation) and the OA:SO₄ was more influenced by changes in SO₄ than OA (Figure 3c). The PILS composition was
strongly dominated by nssSO₄²⁻ and SS, with SS: nssSO₄²⁻ spanning 0.4-1.4. PILS SS showed a marked reduction in US, that
was reflected (qualitatively) by proxies for coarse particles (V-LAS>1μm, V-APS, V-FCDP, N_APS, N_FCDP), while concentrations
in the other airmasses were relatively consistent. The aforementioned DN-DS structure seen in dry scattering, V-LAS<1μm,
PILS nssSO₄²⁻ and to a lesser degree AMS SO₄ was not observed in PILS SS (Figure 3d) and SS abundances in the downwind
airmasses may be primarily driven by local regeneration, supported by the higher observed wind speeds. The lower submicron
aerosol mass in DS (as quantified by the sum of the AMS constituents) as well as other mass proxies (e.g., scattering and LAS
sub-micron volume) supports the prior suggestion of increased precipitation influence on this airmass (i.e., in spite of additional
pollution sources), either through direct removal of aerosols or by injection (by evaporatively cooled downdrafts), and
subsequent mixing, of overlying aerosol-depleted layers.


AMS NH₄ indicated low levels of sulfate neutralization (32-34%) with a minor elevation in DS (53%), consistent with the
insinuation of increased terrestrial influence. Particularly striking was the complete absence of detectable PILS NH₄⁺ in all
four airmasses, whose only constituents were SS and nssSO₄²⁻ and a very minor contribution from NO₃⁻ (assumed to be
associated with the SS given the otherwise acidic conditions). It is unlikely that NH₄⁺ was actually present in the samples, but
not measured during the lab analysis, because the routinely-interspersed standard solutions showed no anomalies in




quantification of NH$_4^+$. Further, the independent PILS-Conductivity measurement shows strong closure with the SS and $_{nss}$SO$_4^{2-}$ under the assumption of balance by H$^+$ (Figure 3e) for the downwind survey – this is notable because of the strong charge-equivalent conductivity of H$^+$ (approximately five times higher than NH$_4^+$), such that closure would not be achieved with other cations. Although NH$_4^+$ fractional losses due to volatilization have been documented for the PILS (Sorooshian et al., 2006), this would not explain complete loss, especially under acidic initial conditions. Barring any further explanation for the PILS, it is possible that sulfuric acid particles in the humid tropical boundary layer retain sufficient water in the AMS to challenge the standard fragmentation assumptions (Allan et al., 2004) producing a positive AMS NH$_4$ artifact, due to water interference. The ratio of AMS SO$_4$ to PILS $_{nss}$SO$_4^{2-}$ was found to be 1.02-1.08 for airmasses except DS, consistent with high AMS SO$_4$ collection efficiency (CE) (Zorn et al., 2008). Curiously, the ratio was 1.6 for the DS airmass (which also had higher AMS NH$_4$) but the fractional change in the PILS mass compared to mass proxies (e.g., scattering and V-LAS$_{<1\mu m}$) between DN and DS showed closer alignment (Figure 3). In this environment, where processing of particles by clouds is almost guaranteed, some degree of internal mixing between SS and $_{nss}$SO$_4^{2-}$ may be expected. In a thermodynamic modelling study, Fridlind and Jacobson (2000) showed that such a mixing state may exert influence over the NH$_3$(g)-NH$_4^+$ partitioning in remote marine environments. This does not explain the AMS-PILS discrepancy; however it highlights a need for further laboratory investigation to determine instrument performance for acidic sulfate under high SS loading, since this pertains to a large range of marine conditions, both background and polluted.

During Case II, low-level sampling occurred upon completion of the CM extending initially to the south of the main line of convection followed by a transect under the convective line and a general survey pattern to the north and west (spanning approximately 180 km), in the vicinity of the R/V Sally Ride. Time series (equivalent to Figure 3) are shown in Figure 4 for this period. Aerosol and trace gas concentrations were very steady with the exception of the two periods highlighted grey, the latter of which corresponded to the time spent sampling to the south of the cloud line (see Figure 2b, SOUTH), while the former period was encountered immediately after exiting a region of precipitation near cloud base at the upwind (i.e., northeast) end of the cloud line. Even through no altitude change occurred, a rapid increase in potential temperature (θ) was observed (~1K) as shallow cumulus on the periphery of the main convective line quickly dispersed. This layer was thermodynamically just above the (locally cloud-free) boundary layer in the lower free troposphere (LFT) and was accompanied by a marked reduction in water vapor (q$_v$), CO, CH$_4$, and aerosols. Similar vertical structure and properties were observed during the spiral climb at the end of the low-level survey (approximately 2 hr later and 200 km to the west) and we interpret LFT as a representative reference state of the broader lower free troposphere, although it cannot be definitively stated whether the sharpness of the gradients atop the mixed layer were reflective of the larger scale background (and upwind) conditions or a dynamic response to the nearby convection. All other periods of the low-level survey (BKGD) were conducted within the PBL and occurred mainly north and west of the convective line in a region that was characterized by a mixture of cloud-free regions and shallow, non-precipitating cumulus clouds and, based on Figure 4, was free of underlying larger-scale gradients



that may confound any interpretation of cloud/precipitation effects on aerosols – we consider this airmass the unperturbed
background. A summary of the airmass properties for BKGD, LFT and SOUTH can be found in Table 3.

SOUTH represents a perturbation of the near-surface environment compared to BKGD with approximately 15-20% reduction
in aerosol (as quantified by scattering, V-LAS$_{<1\mu m}$, and SO$_4$ measurements). The extent to which the properties of SOUTH
could be explained by entrainment and subsequent mixing of LFT into BKGD was evaluated by calculating a putative mixing
fraction ($\chi = \frac{SOUTH-BKGD}{LFT-BKGD}$) representing, for each property, the required proportions of each airmass. Long-lived, passive trace
gases (CO and CH$_4$) imply $\chi \approx 0.4$ and were regarded as the most suitable given that net local sources were unlikely. CO$_2$ also
showed broad agreement ($\chi=0.32$), but the dynamic range was small. A large number of airmass properties that characterize
sub-micrometer aerosol (AMS SO$_4$, PILS $_{nss}$SO$_4^{2-}$, V-LAS$_{<1\mu m}$, dry scattering, all CN, N$_{LAS}$) indicated mixing fractions in the
range 0.38-0.52, suggesting that the downward mixing implied by the trace gases could also explain the reduction in these
aerosol properties. A second metric was computed, $\varepsilon_{0.4}$, defined as the observed anomaly in SOUTH relative that expected for
$\chi=0.4$, (based on the mixing fraction of CO). Based on positive anomalies, PILS SS and corresponding coarse mode proxies
(V-LAS$_{>1\mu m}$, V-APS, V-FCDP, N$_{FCDP}$, N$_{APS}$) all indicated a net coarse particle source in SOUTH, with many unexplainable by
mixing in any proportion, while O$_3$ and OA (marginal) indicated a net sink. The degree of mixing and the location of SOUTH
on the down-shear flank of the line of convection, suggests that evaporative cooling of precipitation falling through the dry
LFT airmass could have been the driver for its downward transport. The thermodynamic variables ($\theta$, q$_v$, $\theta_E$) indicate
moistening and cooling, but a net source of $\theta_E$ (conserved under evaporation of hydrometeors) likely implies an additional
contribution from surface fluxes. SOUTH cannot be considered a "cold pool", since it is not cold; however, it displays the
hallmarks of an airmass that has been affected by penetrative downdraft induced mixing (in the recent past), followed by a
surface flux-driven recovery perhaps accelerated by surface gustiness. The collective net effect being a higher surface layer
$\theta_E$, coarse aerosol enhancement from sea spray fluxes, surface O$_3$ uptake but, crucially, only marginal net losses observed in
the sub-micrometer aerosol budget.

Aerosol composition was similar to Case I, specifically the PILS composition was dominated by SS and $_{nss}$SO$_4^{2-}$ (with a lower
SS:$_{nss}$SO$_4^{2-}$ of 0.31-0.39) and AMS SO$_4$ was the dominant species, with the slightly higher OA:SO$_4$ (ranging 0.28-0.32) with
similar OA age marker ($f_{44}$=0.21). Comparable to Case I, the PILS samples were absent NH$_4^+$, while the AMS NH$_4$ indicated
56-67% sulfate neutralization (i.e., indicative of ammonium bisulfate). Ratios between AMS SO$_4$ and PILS $_{nss}$SO$_4^{2-}$ were 1.25-
1.30 suggesting high CE but also leaving open the possibility of contributions to AMS SO$_4$ from non-refractory species not
detectible by PILS as $_{nss}$SO$_4^{2-}$, such as organosulfates (Farmer et al., 2010; Surratt et al., 2007). Although the PILS composition
was similar to Case I, the conductivity closure afforded by Case II (Figure 4e) was not as complete, with the estimated H$^+$
required for charge balance exceeding the total ion conductivity by 48%. For reference, replacing the H$^+$ with the equivalent
molar concentration of NH$_4^+$ to neutralize the sulfate would underestimate the total ion conductivity by 26%; alternatively, a



65% $NH_4^+$ / 35% $H^+$ mixture would achieve optimal closure (assuming no other contributing constituents). This level of neutralization supports the AMS data suggesting bisulfate but, if correct, does not provide an explanation for underprediction by offline IC.


Case III provided an opportunity to investigate both (i) the variability in aerosol composition attributable to source heterogeneity for a major biomass burning transport event over the Sulu Sea, and (ii) the downwind evolution. A time series of the biomass burning plume in situ transect (Figure 5) illustrates the structure of the plume approximately on a cross wind axis from southeast to northwest and within the plume, concentrations were highly variable both in the vertical and horizontal.

A series of short, stacked legs were flown close to the end points of the transect with substantial aerosol enhancements (dry scattering >200 $Mm^{-1}$) extending up to around 2.5 km. The vertical structure of the plume was not consistent between the two sets of stacked legs, nor was there a monotonic decrease in aerosol abundance with height. Analysis of the upwind airmass history (not shown) provides strong support that fires in southern Kalimantan are the source of the smoke with an age of between 48 and 72 hours. Despite the potential for source variability across the transect, differences in secondary aerosol

production, and variable influence of prior cloud processing, there is a close correspondence in the plume structure amongst CO, dry scattering, LAS volume and OA, which is the largest component of the submicron aerosol mass.

Oxalate, $_{nss}K^+$, and $_{nss}Ca^{2+}$ were found to be enhanced in the plume (Figure 5c), consistent with other studies of biomass burning (Andreae, 1983; Yamasoe et al., 2000; Maudlin et al., 2015); however, the relationship between these ion tracers and OA was

more variable. Oxalate initially tracked OA during the southeast set of stacked legs (0030-0100 UTC) comprising approximately 1.5% of the OA mass, then remained fairly constant despite an increase in OA through the central section of the transect (dropping to ~0.7% of OA) before rebounding near the northwestern section (~0130 UTC). During the subsequent stacked elevated legs oxalate broadly tracked OA, except during the latter half of the highest leg where it remained constant despite rising OA. $_{nss}K^+$ was generally lower at the southeast end of the transect relative to OA, potentially as a result of fuel

differences or mechanisms that resulted in higher loss of primary aerosol in that section of the plume. $_{nss}K^+$ and (more significantly) $_{nss}Ca^{2+}$ exhibited periodic high anomalies (e.g., 0106 UTC, 0130 UTC) with no temporal correlation to other data shown in Figure 5 or other coarse proxies (e.g., V-APS, V-FCDP). A possible explanation relates to the performance of the PILS when sampling coarse and insoluble (or low solubility) particles such as ash and/or crustal material associated with biomass burning. In the PILS, droplet growth and separation from the airstream as an aqueous solution is relatively well

constrained for soluble material; however, Wonaschütz et al. (2018) describe accumulation of insoluble material on the wicking material and impactor in association with laboratory experiments performed with soot particles, while Orsini et al. (2003) also noted similar limitations for low solubility calcium salts. We hypothesize that under constant rinsing, and particularly under varying environmental conditions, insoluble deposits may leach, or become detached, back into the PILS liquid stream in a process that is unlikely to be steady or controlled, perhaps explaining the intermittent structure. Other explanations, such as

artifacts introduced during offline analysis, are less likely since these spike enhancements are significantly higher than



variability observed in sample blanks and were not observed in other phases of flight outside of the biomass burning plume. While this does not help to reconcile the fine scale temporal structure of $_{nss}Ca^{2+}$ (as a proxy for crustal material associated with the biomass burning), the quantification was assumed to be suitable for assessing average plume properties, on the basis that spike enhancements did not continue after leaving the plume. Associated with the $_{nss}Ca^{2+}$ spikes were positive anomalies in

$NO_3^-$ and $NH_4^+$, which otherwise showed very close agreement between AMS $NO_3$ and $NH_4$ (Figure 5e-f). This pattern may be indicative of uptake of $NO_3^-$ and $NH_4^+$ on dust particles associated with the smoke and it provides an explanation for the large departure of $NO_3^-$ at 0130 UTC.

Comparison of $_{nss}SO_4^{2-}$ with the AMS $SO_4$ shows strong coherence across the plume, with a ratio 0.70-0.74 (Figure 5d). In

contrast to Cases I and II, high neutralization was observed by both PILS $NH_4^+$ and AMS $NH_4$ suggesting ammonium sulfate with evidence of $NO_3^-$ contributions to accumulation mode particles, supported by AMS $NO_3$. AMS $SO_4$ exceeded the known AMS CE=0.5 for pure ammonium sulfate particles (Middlebrook et al., 2012), which may be caused by the large OA component. While a CE < 1 likely also applies to the AMS $NH_4$, we expect a $NH_4^+$ volatilization loss for PILS for neutralized ammonium sulfate particles (Sorooshian et al. 2006) explaining the closer AMS $NH_4$ – PILS $NH_4^+$ agreement and larger

difference for sulfate (i.e., where only AMS has reduced CE, compared to Cases I and II). In stark contrast to the other cases, the measured ions (and associated $H^+$ estimate) only explain about half of the measured conductivity (Figure 5g); an expected result given the large number of organic anions (e.g., associated with carboxylic acids) that are not quantified during IC analysis. The structure of the measured conductivity tracks the previously described proxies for the plume (CO, dry scattering, LAS volume, OA) and captures some of the finer scale features smoothed out by the discrete PILS samples (e.g., 0210-0215

UTC). While there is no means to attribute the residual fraction of the conductivity, the fact that it tracks OA may imply that, collectively, carboxylic acids contribute a relatively constant fraction of OA, despite the relative contribution of individual species (as illustrated by oxalate) being somewhat more variable, and concurs with a consistent $f_{44}$=0.14 across the plume.

The CM for Case III, was located about 250 km downwind of the transect. While not a true Lagrangian comparison, it does

provide a point of comparison within the plume subject to approximately eight additional hours of downwind transport. In addition, while small shallow cumulus were present along the upwind transect (and satellite imagery confirmed the presence of shallow cumulus interspersed throughout the environs of the plume's transport history over water), the region of the Sulu Sea near and to the north of the transect appeared to be the first contact for the smoke-laden airmass with more vertically-developed maritime cumulus convection. Trajectory analysis taken at the CM location confirmed that the upwind transect

(TX) represented an upwind condition; however, the observed confluence of the wind streamlines (Figure 2c) made further dissection of the region of the plume somewhat uncertain. Trajectory sensitivity in the vertical, temporal and horizontal (see Figure S3a for details) confirms this and we demark subsections of the plume transect originating from the likely southeast (TX-SE) and northwest (TX-NW) bounds (which happen to capture, in part, the heterogeneity of TX) to evaluate sensitivity to the upwind origin. In lieu of additional information about the temporal steadiness of the advected airmass properties, these





bounds also could capture uncertainty with the temporal mismatch. Ratios of gas tracers (CO, $CO_2$, $CH_4$) provide some additional guidance (see Fig S3b,c) and likely indicate that CM is not fed exclusively with TX-SE. TX-NW alone cannot be rejected, but the most plausible given the expected transport is that the plume becomes more homogenized by the CM and TX mean conditions are probably representative. Trajectories across a longer 5-day timeframe (Figure S4), further highlight the bifurcation caused by the flow pattern around Borneo, which is a likely driver of the varying plume properties across TX. In

contrast, Cases I and II airmass distinctions appear more locally driven than influenced by disparate origin at the larger scale.

Mean properties of CM, TX, TX-SE, TX-NW can be found in Table 4. The percentage change, $\Delta$, from TX to CM was evaluated by using CO as a dilution tracer, with a 0.1 $ppm_v$ background (i.e., $\Delta = (X_{CM}/\Delta CO_{CM})/(X_{TX}/\Delta CO_{TX})$ for any species X and where $\Delta CO = CO - 0.1$). Given the considerably elevated CO, the result is quite insensitive to the choice of

background. The reported evolution is plausible given the expectation of continued plume aging, specifically, 5-10% increase in AMS $SO_4$, $NO_3$ and $NH_4$ and PILS $_{nss}SO_4^{2-}$ in contrast to reductions observed in primary biomass burning aerosol tracers ($_{nss}K^+$ and $_{nss}Ca^{2+}$) and, notably, OA indicates no net change, while oxalate was significantly reduced. Microphysically, a (very) minor decrease in CN is observed along with an increase in accumulation mode number ($N_{LAS}$), consistent with coagulation and the further addition of secondary aerosol. Coarse aerosol evolution shows the combined effects of depositional losses of

primary biomass burning particles in conjunction with significant addition of SS during passage across the Sulu Sea.

### 4.2 Non-sea-salt sulfate cloud-aerosol mass closure

$_{nss}SO_4^{2-}$ is a useful candidate for closure analysis because it is abundant across the three cases, it has a low volatility in all

forms (thereby negating gas phase contributions), and is reasonable to assume that the majority of the particle mass is found in the sub-micrometer size range (i.e., making it readily quantifiable by the airborne in situ aerosol measurements). A PILS-CW $_{nss}SO_4^{2-}$ comparison provides the most consistency but the AMS offers the benefit of improved time resolution, which is valuable for evaluating the clear air vertical profile and for avoidance of cloud/rain contamination. The AMS $SO_4$ was scaled by a case-specific PILS-AMS ratio, under the assumption that the ratio remains near constant at all levels. CW $_{nss}SO_4^{2-}$ (AEM,

referenced to standard density) reflects one component of the total $_{nss}SO_4^{2-}$ in a cloudy parcel, the other being the interstitial fraction. Comparison of CW $_{nss}SO_4^{2-}$ with clear column data is made with respect to altitude (Figure 6a-c) and CO (Figure 6d-f), as a conserved tracer.

Availability of clear column data within the CM vary between cases and Case I, in particular, is mostly reliant on auxiliary

data from nearby profiles, because of the cloud extent. In all cases, clear column $_{nss}SO_4^{2-}$ shows a general decrease with altitude through the cloud layer. In Cases I and II, the decrease is monotonic, while in Case III, there is a minor enhancement in the upper third of the cloud layer reflecting the ongoing adjustment of the clear column profile to recent (and nearby) convective mixing (i.e., through convective detrainment). Positive curvature in the clear column relationship with CO (Figure 6d,e) is



indicative of net $_{nss}SO_4^{2-}$ loss due to precipitation, with more widespread and intense precipitation in Case I resulting in greater
curvature. Case III (Figure 6f), an environment relatively unaffected by precipitation, shows slight negative curvature suggesting net $_{nss}SO_4^{2-}$ production (i.e., $_{nss}SO_4^{2-}$ concentrations within the cloud layer are higher than mixing alone would indicate).

CW $_{nss}SO_4^{2-}$ is generally bounded by the mixed layer $_{nss}SO_4^{2-}$ and, with increasing precipitation (i.e., Case I), the variability
increases, although there is no dominant altitude trend nor alignment with the clear column profile (Figure 6a-c). A greater number of CW samples exceed the clear column $_{nss}SO_4^{2-}$ at an equivalent altitude consistent with upward transport of sub-cloud air by convection, but conditions where the CW $_{nss}SO_4^{2-}$ is lower than the environment are not uncommon (particularly in Case I) suggesting cloudy regions that were either heavily affected by rainout, or where mixing with the environment has significantly reduced the scavenged fraction. The first CW sample in Case II was removed, since the combination with the
LWC resulted in anomalously high AEM concentrations, despite the relative composition being relatively consistent with other samples.

Each CW sample can be normalized by a sub-cloud reference $_{nss}SO_4^{2-}$ (using airmasses: Case I: UN; Case II: BKGD; Case III: CM), an adiabatic LWC, and a reference $N_d$ (using the 90th percentile, as a broad estimate of typical conditions at initial
activation), thereby assessing the level of deviation from an idealized, undilute ascent (Figure 7). A pattern emerges where CW $_{nss}SO_4^{2-}$ generally exceeds that implied by reductions in condensate from the undilute parcel implying that rainout alone is not driving the variability amongst samples. The comparison between normalized CW $_{nss}SO_4^{2-}$ and $N_d$ (Figure 7b) suggests that, when neglecting the highest RWF instances (i.e., indicative of unsaturated precipitation samples), microphysical processes are potentially exerting the dominant control over the intra-cloud $_{nss}SO_4^{2-}$ abundance, compared to other factors such
as chemical production. The singular Case III data point below the 1:1 line in Figure 7a is perhaps indicative of the fact that, in this case, a reference parcel with 100% scavenged fraction is unrealistic, even under undilute conditions, given the competition for water vapor in the highly particle-loaded smoke plume.

Considerable scatter exists in attempts to relate normalized CW $_{nss}SO_4^{2-}$ to other controlling variables (Figure 8) with little, if
any, statistical correlation. The data are grouped in quartiles to explore mean trends and potential influences. RWF and (normalized) altitude exhibit codependence so emergent patterns may be connected (Figure 8a,b). The general altitude trend, although weak, is for a decrease with altitude through the lower 50-70% of the cloud followed by a reversal in the uppermost region. Even though the quartiles are dominated by the data associated with Cases I and II, the three upper CW samples in Case III also show this gradient reversal. A potential explanation is that emergent convective cloud top regions are heavily
biased towards energetic parcels that may be less dilute, while mid-cloud regions, although still containing energetic cores, also comprise older, more mixed and, especially in Case I, detrained stratiform cloud. The RWF dependence offers a similar pattern, but a key finding is that for CW $_{nss}SO_4^{2-}$ no change in AEM is found for cloud-free precipitation compared to non-





precipitating clouds. CW samples collected in updrafts are, all else equal, more likely to contain higher CW $_{nss}SO_4^{2-}$ (Figure 8c), in line with expected convective mass flux of sub-cloud air and the expectation that updrafts would tend to retain a higher
scavenged fraction than downdrafts.

### 4.3 Cloud Composition

#### 4.3.1 Nitrate and Sea Salt

Unlike $_{nss}SO_4^{2-}$, where we expect the sub-cloud aerosol measurements (i.e., PILS and AMS) to capture the majority of the total abundance, total aerosol SS is expected to be grossly underpredicted because of sampling constraints above the inlet 4 μm cut-point, making it challenging to directly compare aerosol to the CW SS. CW $NO_3^-$ has a multitude of potential sources (e.g., Prabhakar et al., 2014); however, we can consider three groups: (i) accumulation mode aerosol activation (e.g., as part of a
sulfate-ammonium-nitrate-organic mixture), (ii) coarse mode aerosol mainly in association with uptake on SS, but also potentially from dust; and, (iii) gas-phase partitioning of nitric acid within cloud (including reactions that produce nitric acid).

Figure 9a summarizes the relationships amongst $NO_3^-$, $_{nss}SO_4^{2-}$, and SS for CW and sub-cloud aerosol from the PILS. PILS SS:$_{nss}SO_4^{2-}$ underpredicts the range spanned by CW (as anticipated) and Case III has a lower SS:$_{nss}SO_4^{2-}$ in both sub-cloud
aerosol and CW – and in alignment with this case being heavily polluted by smoke. The variability seen in CW SS:$_{nss}SO_4^{2-}$ can generally be attributed to some combination of the effects of $_{nss}SO_4^{2-}$ in-cloud production, differential scavenging and precipitation loss mechanisms and, potentially, highly localized variability in sea spray. Notably, this range is substantial for Case I and decreases in parallel with the decrease in precipitation observed between the cases, with Case III exhibiting relatively invariant CW SS:$_{nss}SO_4^{2-}$. CW $NO_3^-$:$_{nss}SO_4^{2-}$ is similarly variable, but there is a much stronger relationship between
$NO_3^-$ and SS on an intra-case basis; increasing in ratio from Case I (0.18) to Case III (0.72). Such a tight relationship between CW $NO_3^-$ and SS in Cases I and II is likely indicative that a major fraction of the CW $NO_3^-$ is already associated with coarse SS upon activation, while there are indications of additional nitric acid partitioning with altitude (Figure 9b) in the polluted, largely non-precipitating, Case III. Case III also has a CW $NO_3^-$ contribution from (previously discussed) accumulation mode particles (Figure 5), yielding a higher PILS $NO_3^-$:SS (1.15) than CW (0.72). The opposite is seen in Cases I and II, suggesting
that the fraction of SS measured by the PILS (i.e., particle diameter < ~4 μm) contain less $NO_3^-$ than the aggregate SS found in CW, and may point to an increased degree of internal mixing of SS and $_{nss}SO_4^{2-}$ on smaller sea salt particles (e.g., Sievering et al., 1990).

The trend in CW SS: $_{nss}SO_4^{2-}$ is downward with (normalized) altitude (Figure 9c) and upward with RWF (Figure 9d) for Cases
I and II. While acknowledging relatively modest correlation, Case I rainwater (RWF>0.9) exhibits higher CW SS: $_{nss}SO_4^{2-}$ (4.24) than non-precipitating (RWF<0.1) regions of the cloud system (2.39; i.e., an increase of 78%) and may be indicative of





a "giant CCN effect" (e.g., Feingold et al., 1999) where large SS particles are responsible for larger cloud droplets thereby increasing the likelihood of both acting as precipitation embryos or being collected by existing rainwater. Additionally, large SS is more susceptible to washout than accumulation-mode $_{nss}SO_4^{2-}$, so progressive enrichment of CW SS in unsaturated 670 environments may be a contributing factor.

Numerous CW samples (20) were collected during two mid-cloud legs of the CM in Case I (Figure 10), in near-continuous sampling conditions. Four convective periods (marked in Figure 10a) were identified from enhancements in the w variability that coincided with local LWC maxima, with other periods representing the more quiescent stratiform clouds and (non-cloudy) 675 precipitating regions. CW SS and $_{nss}SO_4^{2-}$ are locally enhanced in convective periods 1, 3 and 4, coinciding with instances where local enhancement in $N_d$ was observed indicating fresh lofting of sub-cloud air. The lower (2nd) leg shows a higher CW SS:$_{nss}SO_4^{2-}$ overall and is more dominated by high RWF conditions, and there is a consistent trend in both legs for a downwind decrease in CW SS:$_{nss}SO_4^{2-}$ in line with claimed enhanced removal of SS (but still does not explicitly rule out contribution from in cloud $_{nss}SO_4^{2-}$ production). The recovery from any preferential SS removal appears to be rapid in this sub-cloud 680 environment based on the reported characteristics of airmass DS; a region observed to be no longer experiencing, but recently affected by, widespread precipitation but through the action of locally increased wind speed had the highest PILS SS: $_{nss}SO_4^{2-}$ of the four quadrants (Table 2).

SS subjected to aqueous production (e.g., Alexander et al., 2005; Chameides and Stelson, 1992; Sievering et al., 1992), or 685 condensation, of acidic vapors (e.g., nitrate, sulfate and organic acids) tends to liberate Cl$^-$ in the form of gaseous HCl (Seinfeld and Pandis, 2016) resulting in a Cl$^-$:Na$^+$ mass ratio that is typically lower than sea water (1.8). Analysis of the aerosol and CW ratios (Figure 9f) shows evidence of Cl$^-$-depletion, with a stronger effect occurring for the aerosol in each case. CW Cl$^-$-depletion increases in line with more CW NO$_3^-$, while the aerosol-cloud difference may be explained by the PILS bias towards small SS particles, which aligns with observed size dependence of Cl$^-$-depletion elsewhere (Yao et al., 2003; Keene and Savoie, 690 1998) and the potential for repartitioning of HCl in the more dilute CW. Excess aerosol Cl-depletion (i.e., departure from 1:1) is high for Cases I and II (acidic sulfate), where $_{nss}SO_4^{2-}$ likely condensed/coagulated or formed on small SS, and low for Case III (neutralized) where small SS particles remain (relatively) externally mixed from the smoke particles. The level of disequilibrium may be greater in Case III, where SS is a recent addition to the aerosol loading. We can observe a decrease in PILS Cl$^-$:Na$^+$ from TX to CM in line with aging and equilibration of the SS to the abundant acidic gases expected in the smoke, 695 despite some increase in the overall SS (Table 4), while the removal by precipitation in Case I followed by rapid regeneration in the downwind airmasses (e.g., DS) indicates an increase in PILS Cl$^-$:Na$^+$ expected with fresh SS. In summary, Case III may be expected to shift further left in Figure 9f with time, while Case I DS and DN may tend to move slightly left, relaxing towards their upwind counterparts.




A common pattern with altitude is observed (Figure 9e) with CW Cl⁻:Na⁺ increasing in the lower 50-70% of the cloud and then reverting downwards through the upper 30-50%, and the two Case I high-NO₃⁻ outliers correspond to the highest Cl⁻:Na⁺, while the two Case I high-SS outliers correspond to the lowest Cl⁻:Na⁺ of that case. Reasons behind the shape of the profile are not entirely conclusive, but the consistency between cases is certainly notable. Regions of the cloud systems with the highest Cl⁻:Na⁺ align with regions where the aggregate cloud contact time might be longest, suggesting either a relaxation time associated with HCl repartitioning (Keene et al., 1986) could be an important contributor or a timescale associated with homogenization of droplet chemistry due to collision/coalescence; however this falls short when considering the tight grouping of Cl⁻:Na⁺ at each vertical level – much tighter than the expected spectrum of cloud parcel ages – while microphysical explanations ought to be less influential on Case III. Differences in temperature sensitivity to gas-CW partitioning amongst the spectrum of contributing semi-volatiles may be a potential factor. For Case I, the altitude dependence can be seen (Figure 10b) across the two high-density sampling legs, where higher (lower) Cl⁻:Na⁺ coincides with the lower (higher) SS:$_{nss}$SO₄²⁻ in the upper (lower) leg and the downwind gradient of decreasing SS:$_{nss}$SO₄²⁻ is accompanied by an increase in Cl⁻:Na⁺. Up to this point, in cloud $_{nss}$SO₄²⁻ production has not been ruled out as a potential driver for the change in ratio; however, all else equal, this would be expected to drive more Cl⁻-depletion, not less. Instead, precipitation favors enhanced removal of SS (i.e., with lower Cl⁻:Na⁺, driven lower by washout of aerosol SS) and the airmasses left behind are proportionally enriched in HCl. This is further reinforced by considering the outlier pairs: when removal of low-volatility material is significant (high-NO₃⁻ pair) the Cl⁻:Na⁺ is enhanced (in this case even higher than the sea water ratio), while in cases influenced by anomalous additional sea salt aerosol (high-SS pair) the Cl⁻:Na⁺ more closely resembles the PILS Cl⁻:Na⁺. In the convective periods, Cl⁻:Na⁺ was minimally affected (Figure 10b).

4.3.2 Ammonium

CW NH₄⁺:$_{nss}$SO₄²⁻ increases from Case I to III (Figure 11), a pattern mirrored by the out-of-cloud aerosol data measured by the AMS and (at least for Case III) the PILS. The CW vertical structure is relatively consistent between cases with a maximum in NH₄⁺:$_{nss}$SO₄²⁻ between 30-70% of the normalized altitude very similar to the pattern observed for Cl⁻:Na⁺. Profiles of AMS NH₄:SO₄ were constructed by averaging AMS NH₄ and SO₄ over 10% altitude bins for the lower half and then averaging all data for the upper half – done to mitigate noise in the ratio at low concentration. For Cases I and II, there is no discernable altitude dependence, while Case III shows a reduction only for the upper half bin. Excess aerosol NH₄⁺ in Case III would be expected to demand a significant NH₃ vapor pressure that would be available to dissolve in CW. In all three cases, we can assume that almost all NH₃ dissolves in CW at equilibrium (Quinn et al 1986). Despite not measuring NH₃(g), we can use this fact to qualitatively explain that as the aerosol NH₄⁺:$_{nss}$SO₄²⁻ increases, so too should the positive excess seen in CW NH₄⁺:$_{nss}$SO₄²⁻. One result that still lacks explanation is the apparent disconnect of the PILS NH₄⁺ (or lack thereof) observed in Cases I and II.





In Case I, CW $NH_4^+$ ranges between 0.14-1.3 μg $sm^{-3}$ (equivalent to 0.17-1.63 $ppb_v$ $NH_3(g)$).  If this mixing ratio were to exist in the sub-cloud layer in the presence of sulfuric acid particles, then measurable PILS $NH_4^+$ would be expected.  One possibility is that there is a small background $NH_3$ in the lower troposphere including the cloud layer, but the sub-cloud environment has become extremely ammonia-poor through wet scavenging and surface deposition.  This is consistent with oligotrophic surface waters being strongly deficient in $NH_3$ and $NH_4^+$ but relies on turbulent entrainment being too slow to replenish the layer.  To discount the presence of measurable sub-cloud $NH_4^+$ entirely would suggest an over-estimate by the AMS, thus requiring an explanation (via e.g., a water artifact); however the strong agreement between PILS and microphysical proxies for aerosol mass (e.g., LAS) as well as the internal closure from PILS-conductivity make it challenging to discount the data as an instrument failure.  In a modelling study, Fridland and Jacobson (2000) found that in remote marine environment conditions with an accumulation mode containing equimolar parts $NH_4HSO_4$ and NaCl, the mixing state of particles completely dictated retention (external mixture) or liberation (internal mixture) of $NH_4^+$ and $Cl^-$.  At the time of writing, some of the authors are in the process of designing a laboratory experiment attempting to recreate conditions pertinent to Cases I and II to investigate the effect of (i) sulfate neutralization, (ii) aerosol water and (iii) mixing state with sea salt, on the respective performance of the AMS and PILS.

### 4.3.3 Cloud pH

Most of the CW samples (47 of 50) were analyzed for pH (Figure 12) ranging from 3.82-5.32 with mean 4.41, and while the differences were quite modest, Case III had slightly lower mean pH (4.16).  All cases showed an increase in pH with altitude with the largest change occurring for Case I. An increase in pH is expected under increasing LWC and in the presence of enhanced partitioning of basic gases (e.g., $NH_3$).  In Case II, the high pH samples observed near cloud top also contained enhanced $_{nss}Ca^{2+}$ suggesting influence of entrained dust.  Overall consistency between cases is notable given the differences in composition and aerosol abundance and, while Case I could be expected to exhibit the most acidic aerosol (under typical boundary layer conditions), Case III had the most acidic clouds.  The monotonic profile in pH is also in stark contrast to the more complex pattern seen in $NH_4^+$:$SO_4^{2-}$ and $Cl^-$:$Na^+$, indicating that acidity alone was not the controlling factor on the vertical structure of those ratios.

Also reported (Figure 12b,c) is a charge balance assessing the completeness of the measured ion concentrations. Smaller, semi-transparent markers (Figure 12b) indicate net charge (maximum available) based on the 21 contributing species.  The ambient net charge (large, solid markers) was calculated using dissociation equilibria for each species at the measured pH, which should balance the combined charge from $H^+$, $HCO_3^-$ (dashed curve) plus anything left unaccounted. $HCO_3^-$ was estimated using a fixed $CO_2$ mixing ratio of 400 $ppm_v$; however, its impact on charge for pH < 5 is negligibly small.  Closure for Case I was very good, confirming the simplicity of the composition, while Case II showed an anion deficit at the low and high pH extremes.  In Case III, a sizable fraction of the total anion charge was associated with formate, acetate, and the second



deprotonation of dicarboxylic acids – these weak acids are not effective at lowering the pH to the observed range. Additional unmeasured anions associated with stronger acids ($pK_a < \sim3.5$) would be required to achieve closure. Figure 12c shows the

residual charge (i.e., the difference from the dashed line, accounting for dissociation) normalized against the total measured (sign-independent) as a measure of relative imbalance. Uncertainties associated with ion chromatography yield some cancellation for charge balance when errors are common to anions and cations; however, calibration uncertainties and blank corrections (blanks are not meaningful for pH measurements due to non-linearity and buffering) ultimately limit the interpretability of the residual charge. From this, it can be seen that imperfect closure for Case II appears as a small systematic

offset, likely inseparable from analytical uncertainty, while Case III has 14-19% residual charge that suggests that unmeasured organic acids make up a relatively constant contribution on a fractional basis.

### 4.3.4 Organic closure

TOC was measured for a subset of the CW samples (35 of 50) and compared (Figure 13a) with the eleven speciated organic ions by taking the sum of the organic carbon attributable to each ion (i.e., weighted by species carbon mass fraction), which we describe, for convenience, as organic ion carbon (OIC). The imprint of the biomass burning emissions on the cloud chemistry of Case III is clearly evident through the 5-10 fold increase in both TOC and OIC compared to Cases I and II. A notable result is the consistency of the fraction of TOC explained by OIC, with a regression (total least squares; all cases)

indicating that OIC comprises 25.6% of TOC ($R^2 = 0.81$). Individual OIC:TOC ratios indicate inter-case means that are not statistically different ($p > 0.1$ for any pair of cases based on a Welch t-test) and the finding is similar to Stahl et al. (2021), who conducted a regional survey using the complete CAMP$^2$Ex dataset and found that 70% of the TOC was unaccounted for by ions. For readability in the plot the three points associated with Case III have been scaled by a factor of ten.

Of the eleven speciated organic ions: (i) DMA was not detectable in any of the CW samples (i.e., spanning all cases); (ii) maleate was absent in Cases I and II and only detected in a limited subset of Case III (4 of 6); (iii) in Case III, glycolate was only detectable in one sample – this species elutes near acetate, which was strongly enhanced masking its detectability; (iv) succinate and adipate coelute and are considered in tandem (as succinate) – only Case I contained a few samples (7 of 27) that were analytically judged as adipate; and, (v) MSA was not detectable in half the Case III samples. The major organic ion

contributions were from formate, acetate and oxalate, in varying proportion, with succinate more considerable in Case III (Table S3). CW MSA:$_{nss}SO_4^{2-}$ was 0.26% and 0.43% for Cases I and II, which is close to the lower bound reported by Bates et al. (1992) for submicron particles in biogenically-influenced marine airmasses. They found that in warm tropical maritime conditions, biogenic sulfur precursors (e.g., dimethyl sulfide) do not favor the MSA production pathway (thus making it a poor discriminant for evaluating biogenic sulfur contributions in this scenario), hence on these grounds we cannot negate a (partial)

contribution to $_{nss}SO_4^{2-}$ from marine biogenic origin, but we can conclude that there are no significant sources of MSA (including in-cloud production) affecting these two cases. In Case III, for the three samples where MSA was detected, the ratio



is even lower (< 0.1%) strongly indicating that this smoke plume is not enriched in MSA, at least insomuch as it affects clouds, despite the particulate sulfur content (i.e., $_{nss}SO_4^{2-}$) of the plume being quite high.

In spite of its contribution to organic ions in CW, oxalate was not detected on particles sampled by the PILS in Cases I and II, suggesting that, like $NO_3^-$, oxalate is neither thermodynamically favored on smaller SS particles (that are perhaps enriched in additional $_{nss}SO_4^{2-}$), nor sulfuric acid/ammonium bisulfate particles, externally mixed from SS. Size-resolved aerosol sampling within the region has shown oxalate present on both fine and coarse particles (Cruz et al., 2019) and in the absence of a neutralizing base, oxalate may repartition/volatilize to oxalic acid in the gas phase (Paciga et al., 2014), which may inhibit fine

mode oxalate, compared with the data of Cruz et al. (2019). In a sub-tropical marine environment, Turekian et al. (2003) found an increase in oxalate relative to MSA (i.e., another secondary aerosol tracer), with size, in support of the inclination of oxalate towards coarse SS, but argued that its formation on SS was related to marine organic precursors. Although no supporting gaseous measurement was available, we make the assumption that acetate and formate reside in the gas phase when outside cloud. The partitioning of the other measured carboxylic acids is unknown, but their contributions to TOC closure are small.

AMS OA represents a particle-phase contribution to CW TOC, subject to cloud scavenging. An OA carbon mass fraction ($f_c$) calculated from elemental O:C based on AMS $f_{44}$ (Aiken et al., 2008) scales the OA to allow comparison on a carbon mass basis – assumed to be more constrained than attempting to convert the incomplete CW composition to organic mass. In Cases I and II, the contribution of AMS OA carbon to CW TOC was found to be 48% and 35%, respectively (Figure 13b), an additional 20% and 24% relates to the carbon contribution from acetate, formate and oxalate (assumed not to overlap with

AMS OA), leaving a residual 32% and 40% from water-soluble organic gases (WSOG) – such as alcohols, aldehydes and additional carboxylic acids – and unmeasured coarse mode organic contributions. Comparing Case I and II, the AMS OA:SO$_4$ increases by 78% while CW TOC:$_{nss}SO_4^{2-}$ increases by 120% highlighting a general enhancement in the contribution of organic species to both cloud and aerosol but the increased TOC fraction of acetate and formate is perhaps indicative of the more general influence of WSOG to CW TOC in Case II.


In Case III, the availability of AMS OA carbon was found to exceed CW TOC (Figure 13b). In addition, coarse mode organics and WSOG cannot be discounted as also contributing to CW TOC, but we might expect more overlap in the measured CW organic ions with AMS OA, since the thermodynamic environment is more conducive for these acids to be found (in some proportion) on accumulation mode particles (e.g., because of higher $NH_4^+$). A specific unknown is any potential difference in

CE between AMS OA and AMS SO$_4$, the latter having been reported as 0.7-0.74, based on PILS $_{nss}SO_4^{2-}$, but the former not verifiable. A second explanation for the apparent exceedance of OA is that the scavenged fraction may be lower for the less hygroscopic organic matter compared to sulfate. With such a high number of potential CCN in Case III, strongly selective droplet activation – towards the activation of larger particles – would lead to a higher interstitial mass fraction that could be proportionately biased towards organics, especially if smaller particles tended to be organic-rich. Further, a more detailed

review of the vertical structure (Figure 14) suggests that entrainment would tend to drive further increases in the interstitial



mass fraction through droplet evaporation, especially in the upper half of the cloud where the magnitude of the reduction in CW AEM and $N_d$ (~ 3-5 fold) is far larger than the dilution (as quantified by the reduction in the in-cloud CO above background). The vertical velocity structure observed in the upper half of the cloud (Figure 14b) also indicates that active growth of the current cell has likely ceased with some of the higher LWC regions embedded within downdrafts. This is likely

to favor expulsion of CW AEM thereby enhancing the interstitial fraction, while in weakly recirculating regions reactivation may be even more biased to larger and more hygroscopic particles than during initial activation at cloud base. In contrast, the transect at ~1.3 km comprised a strong (~7 ms$^{-1}$) updraft collocated with the peak LWC, CO enhancements commensurate with the sub-cloud layer and high $N_d$ indicative of a fresh convective element. Despite the stark dynamical and microphysical differences, the organic composition was relatively unchanged, at least as inferred from the speciated organic ions. Overall,

the structure of the AEM concentrations for the carboxylic acids conforms rather closely to that of the inorganic ions and TOC (where measured), with perhaps a minor relative enhancement in formate and acetate in the upper half of the cloud that could be explained in the same manner as nitrate (Figure 9) as an increase in partitioning at lower temperatures. There is no indication of an aqueous chemical mechanism dominating the TOC budget, at least at the time scale of this convective cloud system.

## 5. Conclusions

We have analyzed the sub-cloud and in-cloud composition for three tropical maritime shallow convection cases. These cases were selected from flights carried out as part of the CAMP$^2$Ex field campaign specifically because of their high spatial and temporal coincidence between cloudy and clear air sampling, and the availability of a unique CW composition dataset. In each case, sub-cloud aerosol properties and abundances exhibited mesoscale variability that was attributed to a combination

of the direct influence of the cloud systems on circulation through convergence of distinct airmasses, the impact of vertical mixing and aerosol removal by precipitation, and chemical production effects. Although complicating the cloud closure, sub-cloud airmass variability served as a useful aspect for closure analysis amongst the various aerosol (microphysical and compositional) and gas phase instrumentation. The first two cases could be categorized as having aged, polluted/elevated background conditions, meaning that these airmasses contained a strong signature of elevated East Asian regional pollution

that had resided for several (3-5) days over tropical waters, aging the aerosol properties and largely homogenizing the fingerprint of sources. In contrast, the third case comprised a major smoke transport event from Kalimantan fires. In Cases I and II, the aerosol was dominated by sea salt and nss-sulfate with low levels of neutralization suggestive of a mixture of sulfuric acid and ammonium bisulfate, while Case III was fully neutralized by ammonium with nitrate observed on sub-micrometer particles. The ratio of the AMS $SO_4$ to PILS $_{nss}SO_4^{2-}$ was near (or greater than) unity for the acidic cases, but

reduced to 0.70-0.74 for the smoke case, which can be attributed to reduced CE in the AMS. $NH_4^+$ was not captured equally between the PILS and AMS for the acidic cases, with the PILS suggesting negligible $NH_4^+$ and a relationship with the presence of sea salt believed to be a contributing factor. This ought to be the topic of future investigation under controlled laboratory conditions. The sub-micrometer aerosol in the smoke-enhanced boundary layer of Case III was dominated by organic mass –



a factor of ~6 greater than sulfate. In Cases I and II there was evidence that sea salt was quickly replenished following perturbations related directly to precipitation (Case I) and indirectly through downdrafts and surface gustiness (Case II).

The cloud systems studied in the three cases reflected a decreasing significance of precipitation from Case I to III and, as such, the clear column relationship between CO and sulfate (used as a ubiquitous aerosol tracer) showed a decreasing curvature. High curvature found in Case I indicated widespread vertical mixing of CO accompanied by net loss of sulfate due to 875 precipitation, while slight negative curvature found in the largely non-precipitating Case III was indicative of similar vertical redistribution between species along with net sulfate production. There was a high degree of intra-cloud variability in CW solute abundance which can be largely attributed to physical (rather than chemical) mechanisms, affecting the scavenged fraction (e.g., homogeneous and inhomogeneous mixing of cloud-free air) and losses due to rainout. With the exception of a few outlying samples, CW sulfate was generally bound at the upper level by the sulfate concentration in the sub-cloud layer 880 but could also be found to be lower than the clear column abundance at equivalent altitudes. Acknowledging the sources of measurement uncertainty, the current level of quantitative agreement between aerosol and cloud solute abundances (using a sulfate tracer) is certainly encouraging for these complex and dynamic convective cloud systems; however, further improvement in quantification of scavenged fraction and rainout loss likely requires a technique to accurately measure interstitial mass.


Of the other remaining major species contributing to the CW composition, none could yield an effective closure with sub-cloud aerosol because of large contributions from the under-sampled coarse mode particles and gas phase abundance. In all cases, sea salt was the largest single constituent, by mass, of the average CW composition and implicitly suggested that PILS captured only 10-50% of the total sea salt mass, depending on the case and calculation method. The range of compositional 890 variability between sea salt and sulfate increased from Case III to Case I, in line with an increase in precipitation, and rainwater tended to have higher relative sea salt, which was attributed to (i) a giant CCN effect whereby large sea salt had a greater predisposition to generate large droplets more likely to be involved in warm rain processes, and (ii) a greater capture of sea salt by rainwater outside clouds. Nitrate showed a very tight connection with sea salt in all cases, despite being largely absent in the sub-cloud aerosol measured in Cases I and II. These results imply that the nitrate was already associated with coarse 895 sea salt upon incorporation into cloud, and it was only in Case III where a fractional contribution could be diagnosed from accumulation mode particles (up to 30%) and additional gas-phase partitioning. Similar findings extended to ammonium and organic species with CW concentrations exceeding projections from the measured aerosol species, the one exception being the organic closure in Case III. In this instance, 20-50% of the AMS organic aerosol mass was unaccounted for in CW TOC after consideration of carbon fraction, potential CE differences with AMS SO$_4$, and differences in production rates in cloud between 900 sulfate and organics. This may indicate a higher proportion of organic material resides as interstitial relative to sulfate either through discrimination at activation or through subsequent preferential evaporation as the cloud interacts with dry environmental air. A closure analysis between measured pH and speciated ions indicated that charge neutrality (< 10%) was



achieved in Cases I and II and an estimated 14-19% additional anion contribution (with $pK_a < 3.5$) was needed in Case III. Comparison between the measured CW organic ions (i.e., a subset of carboxylic acids) and TOC indicated that 25% of the

carbon was measured, and this was remarkably consistent between the three cases, despite differences in the relative abundance of the measured organic ions. This result was particularly striking given the order of magnitude difference in concentration between Case III and the other cases, but was supported in part by the relatively consistent and high AMS $f_{44}$ amongst the cases, implying that atmospheric aging, leading to oxidation of organic species, in tropical maritime airmasses is quite rapid and consistent, at least in terms of functionality.


**Data Availability**

All datasets are publicly available and can be found at: hhttps://doi.org/10.5067/Suborbital/CAMP2EX2018/DATA001. Himawari visible imagery were accessed through the CAMP$^2$Ex Worldview Interface: http://geoworldview.ssec.wisc.edu.

**Author Contribution**

EC, LDZ, MAS, CER, ELW, KLT, JPD, GSD, SW, SCV contributed to experimental data collection, EC, RAB, ABM, CS, AS performed laboratory analysis. All authors contributed towards manuscript preparation.

**Competing Interests**

The authors declare that they have no conflict of interest.

**Acknowledgements**

This research was funded by NASA's Radiation Sciences Program and authors from the University of Arizona acknowledge support from NASA grant 80NSSC18K0148. We acknowledge the use of imagery from the NASA CAMP2Ex Worldview
application (https://worldview.earthdata.nasa.gov), part of the NASA Earth Observing System Data and Information System (EOSDIS). We wish to thank the pilots and flight crew of the NASA Wallops Flight Facility P-3 aircraft for their support throughout the CAMP$^2$Ex field campaign. We would also like to recognize the assistance from the NASA Ames Earth Science Project Office (ESPO) and facilities personnel at the Clark International Airport in Pampanga, Philippines. Finally, we would like to thank students and staff at the Manila Observatory and Ateneo de Manila University for help in our field laboratory.

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



l390                              Table 1: Cloud system and environmental summary properties

|  |  |  | Case | | |
|---|---|---|---|---|---|
|  |  |  | I | II | III |
| LCL |  | m | 680 | 840 | 630 |
| Cloud top |  |  | 3500 | 4000 | 4800 |
| LWP |  | g m$^{-2}$ | 1640 | 2920 | 1600 |
| LWC | mean (max) | g m$^{-3}$ | 0.50 (2.85) | 0.69 (5.64) | 0.39 (1.45) |
|  | mean |  | 1.84 | 1.02 | 0.71 |
| P | cloud base | mm hr$^{-1}$ | 3.96 | 1.98 | 0.06 |
|  | peak |  | 5.77 | 3.56 | 2.40 |
| N$_d$ | mean (max) | cm$^{-3}$ | 124 (942) | 334 (1650) | 663 (2100) |
| RH | above cloud | % | 36 | 26 | 41 |
|  | adjacent |  | 81 | 60 | 60 |
| w | max (min) | m s$^{-1}$ | 5.2 (-3.6) | 10.8 (-8.2) | 9.5 (-9.7) |
|  | N$_{CW}$ | - | 27 | 17 | 6 (+1) |
|  | N$_{LEVELS}$ | - | 5 | 7 | 7 |





Table 2: Case I sub-cloud airmass properties pertaining to the upwind north and south (UN, US) and downwind north and south (DN, DS) quadrants surrounding the convective line.

| | | | Case I AIRMASS | | | |
| --- | --- | --- | --- | --- | --- | --- |
| | | | UN | US | DN | DS |
| Met | Z | m | 602 | 329 | 363 | 317 |
| | $\theta$ | K | 300.9 | 299.9 | 300.8 | 300.3 |
| | $q_v$ | g kg$^{-1}$ | 15.8 | 16.0 | 16.6 | 17.0 |
| | $\theta_E$ | K | 347 | 347 | 349 | 350 |
| | $w'_{rms}$ | m s$^{-1}$ | 0.22 | 0.32 | 0.41 | 0.46 |
| | $|U|$ | | 2.7 | 1.6 | 8.3 | 8.6 |
| Trace Gas | CO | | 0.137 | 0.139 | 0.133 | 0.144 |
| | $CO_2$ | ppm$_v$ | 407.9 | 408.6 | 407.8 | 411.8 |
| | $CH_4$ | | 1.950 | 1.950 | 1.947 | 1.967 |
| | $O_3$ | ppb$_v$ | 45.8 | 42.1 | 45.3 | 40.7 |
| AMS | $SO_4$ | µg sm$^{-3}$ | 7.42 | 6.17 | 6.54 | 5.13 |
| | $NH_4$ | | 0.95 | 0.77 | 0.79 | 1.02 |
| | $NO_3$ | | 0.06 | 0.08 | 0.07 | 0.07 |
| | OA | | 1.37 | 1.15 | 1.27 | 1.42 |
| | $f_{44}$ | unitless | 0.22 | 0.18 | 0.17 | 0.19 |
| | OA : SO4 | | 0.18 | 0.19 | 0.19 | 0.28 |
| PILS | SS | µg sm$^{-3}$ | 4.48 | 2.63 | 5.10 | 4.58 |
| | $_{nss}SO_4^{2-}$ | | 6.77 | 6.04 | 6.03 | 3.19 |
| | $NO_3^-$ | | < 0.1 | < 0.1 | 0.11 | 0.16 |
| | SS : $_{nss}SO_4^{2-}$ | unitless | 0.66 | 0.44 | 0.85 | 1.44 |
| SP2 | BC | ng sm$^{-3}$ | 28.1 | 21.7 | 36.4 | 44.8 |
| Mass Proxy | V-LAS (<1µm) | µm$^3$ cm$^{-3}$ | 7.0 | 5.4 | 6.7 | 4.4 |
| | V-LAS (>1µm) | | 5.4 | 3.0 | 5.1 | 4.5 |
| | V-APS | | 38.6 | 22.8 | 47.4 | 43.7 |
| | V-FCDP | | 303.3 | 54.6 | 181.4 | 250.9 |
| | Scat. 550 | Mm$^{-1}$ | 77.7 | 62.8 | 78.6 | 50.4 |
| Micro. | $CN_{>3\,nm}$ | cm$^{-3}$ | 3310 | 2660 | 2885 | 2580 |
| | $CN_{>10\,nm}$ | | 2720 | 2180 | 2377 | 2126 |
| | $CN_{>10\,nm,\,non-vol}$ | | 1210 | 1000 | 1115 | 1096 |
| | $N_{LAS}$ | | 985 | 787 | 879 | 800 |
| | $N_{FCDP}$ | | 1.22 | 0.39 | 0.96 | 1.12 |
| | $N_{APS}$ | | 12.96 | 8.93 | 16.65 | 12.40 |





1400 Table 3: Case II airmass properties pertaining to the sub-cloud background (BKGD), characteristic lower free-troposphere (LFT) and a perturbed sub-cloud airmass located on the southern flank of the convection (SOUTH). $\chi$ and $\varepsilon_{0.4}$ represent mixing fractions and anomalies of the perturbed airmass, respectively (see text).

| | | | Case II AIRMASS | | | | |
| --- | --- | --- | --- | --- | --- | --- | --- |
| | | | BKGD | LFT | SOUTH | $\chi$ | $\varepsilon_{0.4}$ |
| Met | Z | m | 285 | 674 | 149 | | |
| | $\theta$ | K | 300.3 | 301.8 | 300.4 | 0.04 | -0.5 |
| | $q_v$ | g kg$^{-1}$ | 16.6 | 12.1 | 17.4 | -0.17 | +2.6 |
| | $\theta_E$ | K | 349 | 336 | 352 | -0.23 | +8.2 |
| | $w'_{rms}$ | m s$^{-1}$ | 0.34 | 0.25 | 0.29 | | |
| | \|U\| | | 6.4 | 6.2 | 6.2 | | |
| Trace Gas | CO | | 0.167 | 0.142 | 0.157 | 0.40 | - |
| | $CO_2$ | ppm$_v$ | 409.2 | 408.4 | 409.0 | 0.32 | <0.1 |
| | $CH_4$ | | 1.972 | 1.956 | 1.966 | 0.37 | <0.001 |
| | $O_3$ | ppb$_v$ | 38.1 | 33.5 | 31.8 | 1.35 | -4.5 |
| AMS | $SO_4$ | | 6.23 | 4.17 | 5.26 | 0.47 | -0.15 |
| | $NH_4$ | μg sm$^{-3}$ | 1.53 | 0.88 | 1.33 | 0.30 | +0.06 |
| | $NO_3$ | | 0.09 | 0.08 | 0.08 | - | - |
| | OA | | 2.01 | 1.25 | 1.47 | 0.72 | -0.25 |
| | $f_{44}$ | unitless | 0.21 | 0.22 | 0.21 | | |
| | OA : SO4 | | 0.32 | 0.30 | 0.28 | | |
| PILS | SS | | 1.56 | 1.23 | 1.58 | -0.06 | +0.15 |
| | $_{nss}SO_4^{2-}$ | μg sm$^{-3}$ | 4.97 | 3.22 | 4.06 | 0.52 | -0.22 |
| | $NO_3^-$ | | 0.19 | < 0.1 | < 0.1 | - | - |
| | SS : $_{nss}SO_4^{2-}$ | unitless | 0.31 | 0.38 | 0.39 | | |
| SP2 | BC | ng sm$^{-3}$ | 102 | 66 | 90 | 0.31 | +3.1 |
| Mass Proxy | V-LAS (<1μm) | | 5.4 | 3.4 | 4.6 | 0.41 | 0.0 |
| | V-LAS (>1μm) | μm$^3$ cm$^{-3}$ | 2.8 | 2.6 | 2.8 | 0.02 | +0.1 |
| | V-APS | | 17.1 | 11.6 | 21.7 | -0.82 | +6.7 |
| | V-FCDP | | 52.2 | 2.0 | 53.2 | -0.02 | +20.8 |
| | Scat. 550 | Mm$^{-1}$ | 48.7 | 32.9 | 42.7 | 0.38 | +0.2 |
| Micro. | $CN_{>3\,nm}$ | | 2314 | 1557 | 1993 | 0.42 | -22 |
| | $CN_{>10\,nm}$ | | 1906 | 1293 | 1647 | 0.42 | -16 |
| | $CN_{>10\,nm,\ non-vol}$ | cm$^{-3}$ | 1241 | 831 | 1033 | 0.51 | -46 |
| | $N_{LAS}$ | | 1032 | 641 | 855 | 0.45 | -22 |
| | $N_{FCDP}$ | | 0.36 | 0.16 | 0.36 | -0.02 | +0.08 |
| | $N_{APS}$ | | 4.25 | 3.19 | 5.13 | -0.83 | +1.30 |

1405



Table 4: Sub-cloud trace gas and aerosol properties for Case III at the downwind location of the cloud module (CM) and the upwind transect (TX). TX was further divided into southeast (TX-SE) and northwest (TX-NW) segments. Δ is the dilution-adjusted percentage change from TX to CM applied to aerosol measurements.

| | | | Case III AIRMASS | | | | Δ (%) |
|---|---|---|---|---|---|---|---|
| | | | CM | TX | TX-SE | TX-NW | |
| Met | Z | m | 623 | 318 | 312 | 309 | |
| | $\theta$ | K | 302.1 | 300.9 | 300.7 | 301.3 | |
| | $q_v$ | g kg$^{-1}$ | 16.1 | 17.7 | 17.8 | 18.1 | |
| | $\theta_E$ | K | 349 | 353 | 353 | 354 | |
| | $w'_{rms}$ | m s$^{-1}$ | 0.47 | 0.31 | 0.29 | 0.24 | |
| | $|U|$ | | 10.9 | 6.6 | 7.0 | 6.3 | |
| Trace Gas | CO | ppm$_v$ | 0.613 | 0.676 | 0.656 | 0.835 | |
| | $CO_2$ | | 412.9 | 413.6 | 412.3 | 415.3 | |
| | $CH_4$ | | 1.899 | 1.900 | 1.889 | 1.914 | |
| | $O_3$ | ppb$_v$ | 61.9 | 67.7 | 69.6 | 78.7 | |
| AMS | $SO_4$ | µg sm$^{-3}$ | 5.99 | 6.29 | 7.02 | 6.78 | +7 |
| | $NH_4$ | | 3.01 | 3.16 | 3.56 | 3.59 | +7 |
| | $NO_3$ | | 1.13 | 1.22 | 1.18 | 1.60 | +5 |
| | OA | | 35.6 | 40.3 | 35.3 | 54.4 | -1 |
| | $f_{44}$ | unitless | 0.15 | 0.14 | 0.14 | 0.14 | |
| | OA : SO4 | | 5.94 | 6.40 | 5.03 | 8.03 | |
| PILS | SS | µg sm$^{-3}$ | 1.16 | 0.96 | 1.48 | 1.10 | +36 |
| | $_{nss}SO_4^{2-}$ | | 8.29 | 8.47 | 9.65 | 9.74 | +10 |
| | $NH_4^+$ | | 2.61 | 3.32 | 4.00 | 4.29 | -12 |
| | $NO_3^-$ | | 1.33 | 1.15 | 1.36 | 1.79 | +30 |
| | $_{nss}K^+$ | | 0.31 | 0.38 | 0.36 | 0.70 | -7 |
| | $_{nss}Ca^{2+}$ | | 0.09 | 0.12 | 0.09 | 0.41 | -21 |
| | oxalate | | 0.24 | 0.39 | 0.31 | 0.67 | -32 |
| | OA : $_{nss}K^+$ | unitless | 114 | 107 | 98 | 78 | |
| | $_{nss}SO_4^{2-}$ : $_{nss}K^+$ | | 27 | 22 | 27 | 14 | |
| | Oxalate : OA | % | 0.66 | 0.97 | 0.86 | 1.2 | |
| | $_{nss}K^+$ : $\Delta CO$ | µg sm$^{-3}$ ppm$_v$ | 0.61 | 0.65 | 0.65 | 0.95 | |
| Mass Proxy | V-LAS (<1µm) | µm$^3$ cm$^{-3}$ | 41.2 | 43.3 | 45.5 | 56.3 | +7 |
| | V-LAS (>1µm) | | 16.2 | 15.2 | 18.2 | 17.8 | +20 |
| | V-APS | | 31.2 | 39.6 | 48.3 | 39.1 | -11 |
| | V-FCDP | | 38.9 | 57.0 | 98.7 | 26.1 | -23 |
| | Scat. 550 | Mm$^{-1}$ | 586 | 642 | 667 | 812 | +3 |
| Micro. | $CN_{>3\ nm}$ | cm$^{-3}$ | 4001 | 4698 | 4608 | 5521 | -4 |
| | $CN_{>10\ nm}$ | | 3356 | 3924 | 3854 | 4614 | -4 |
| | $CN_{>10\ nm,\ non-vol}$ | | 2426 | 2776 | 2710 | 3315 | -2 |
| | $N_{LAS}$ | | 2481 | 2595 | 2546 | 3310 | +7 |
| | $N_{FCDP}$ | | 0.43 | 0.53 | 0.72 | 0.44 | -11 |
| | $N_{APS}$ | | 20.1 | 25.8 | 31.5 | 26.6 | -12 |



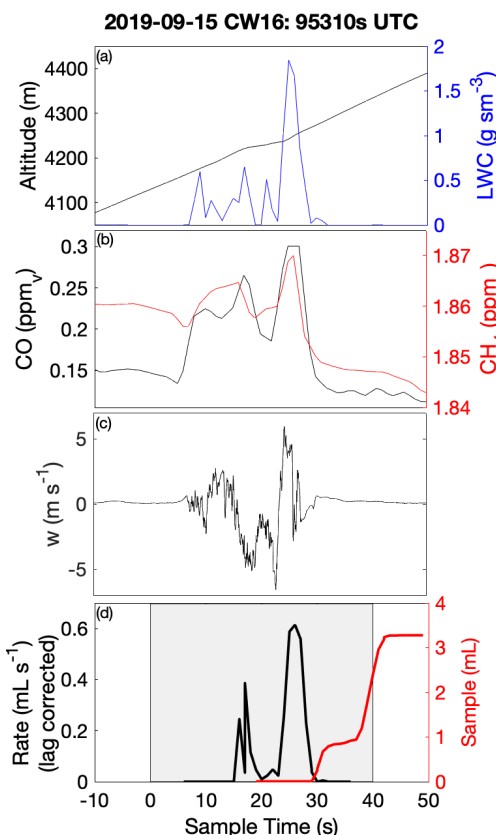

l415

Fig 1: Example CW sample merge with auxiliary data for a convective cloud penetration. Timeseries of (a) altitude and merged LWC, (b) CO and $CH_4$ gas tracers, (c) vertical velocity, and (d) AC3 diagnostic accumulated sample water (right) and estimated lag-corrected collection rate (left). The shaded region indicates the sampling duration, when the AC3 shutter
l420  was open, noting that the physical collection of CW within the system can continue for a short duration thereafter.



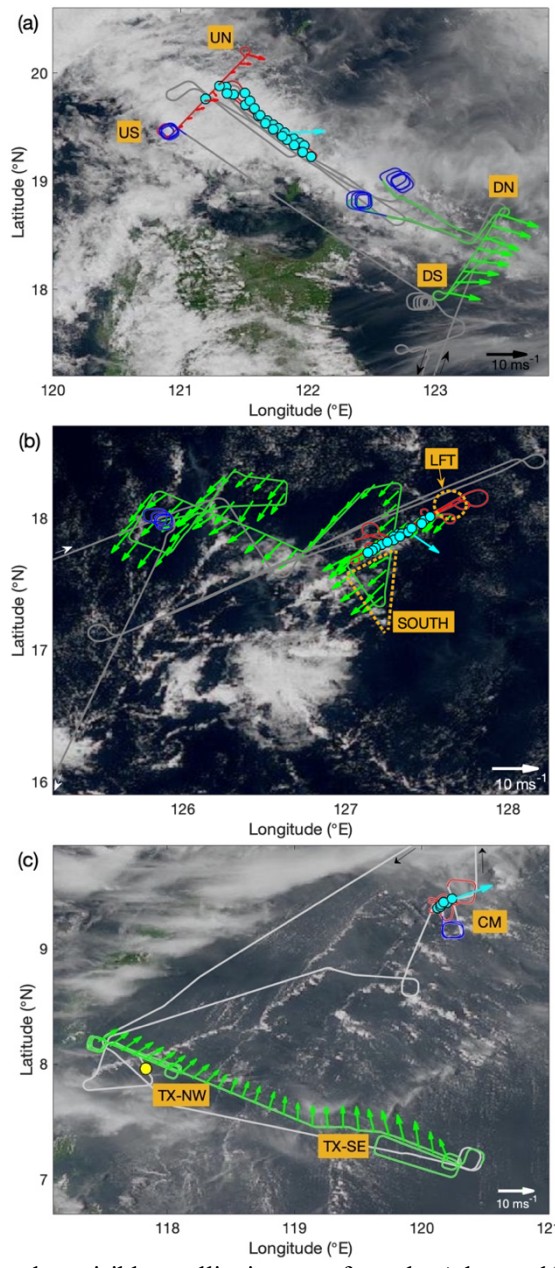

Figure 2: Flight tracks (grey) overlayed on visible satellite imagery from the Advanced Himawari Imager (accessed through the CAMP2Ex Worldview Interface: http://geoworldview.ssec.wisc.edu) taken near the time of cloud sampling for (a) Case I: September 19-20, 2019, (b) Case II: September 23-24, 2019, and (c) Case III: September 15-16, 2019. Green highlights correspond to periods of extended low altitude sampling that were used for characterization of the sub-cloud layer structure, trace gases, and aerosols. Red highlights correspond to the cloud modules: in (a) and (b) these were "wall" patterns with extended level legs at multiple altitudes within, and near to, cloud; in (c) this was a spiral descent with multiple penetrations of cloud at different altitudes. Nearby spiral profiles are highlighted in blue. Cloud water (CW) samples are indicated by cyan-filled circles and in (c) an auxiliary CW sample in yellow. Low-level (green, red) and near cloud top (cyan) wind





vectors are shown using 60 s- and leg-mean horizontal wind data, respectively. Designated airmass locations are shown (see text for details).

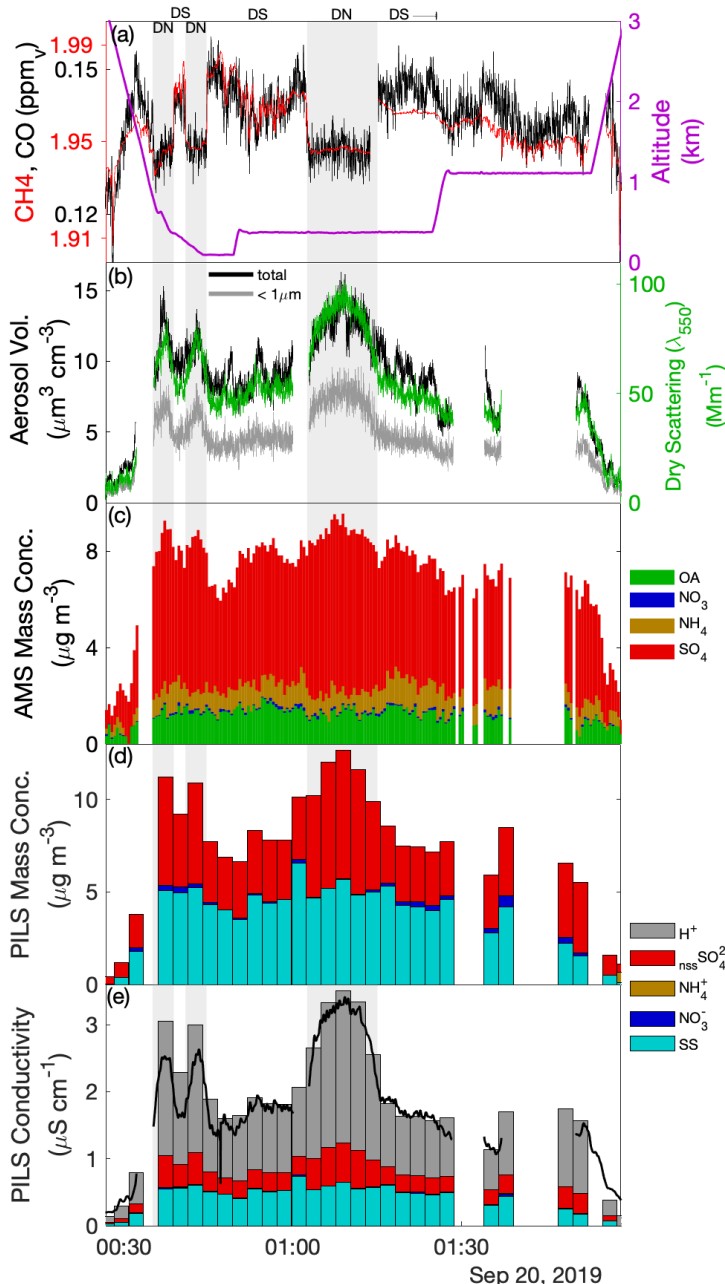

1435 Figure 3: Time series of aerosol and trace gases during low level sampling downwind of the cloud wall during Case I. (a) Gas tracers CO and CH4 and aircraft altitude, (b) Bulk aerosol dry scattering at 550 nm and integrated aerosol volume from the LAS for all size and sub-micrometer size bins, (c) AMS speciated (sub-micrometer) mass concentrations, (d) PILS bulk mass concentrations for major ion groups, (e) PILS bulk conductivity and closure analysis of major ion contributions. H+ ion contributions show the H+ required for charge neutrality. Grey-shaded background regions highlight periods of special 1440 interest (see text).



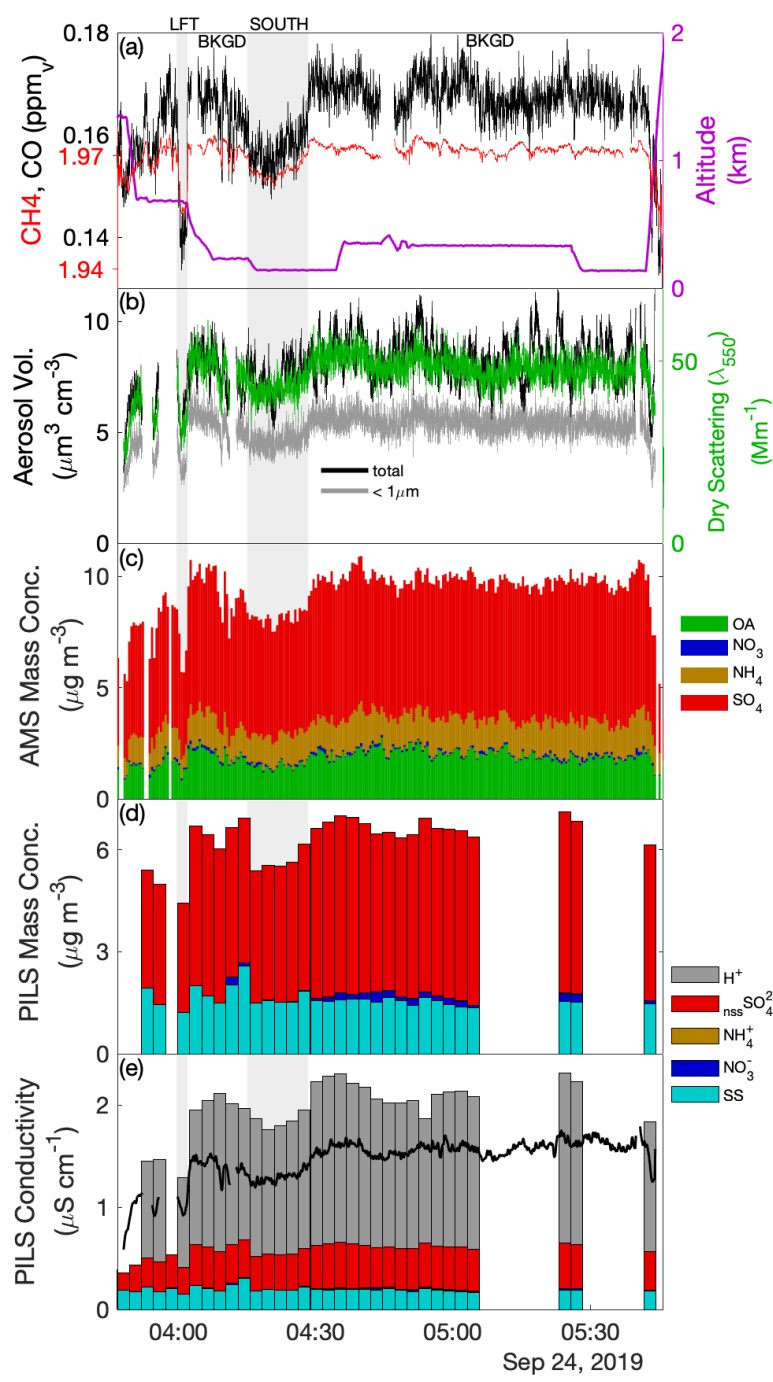

445

Figure 4: As Figure 3 for proximal low-level sampling around the cloud wall of Case II.



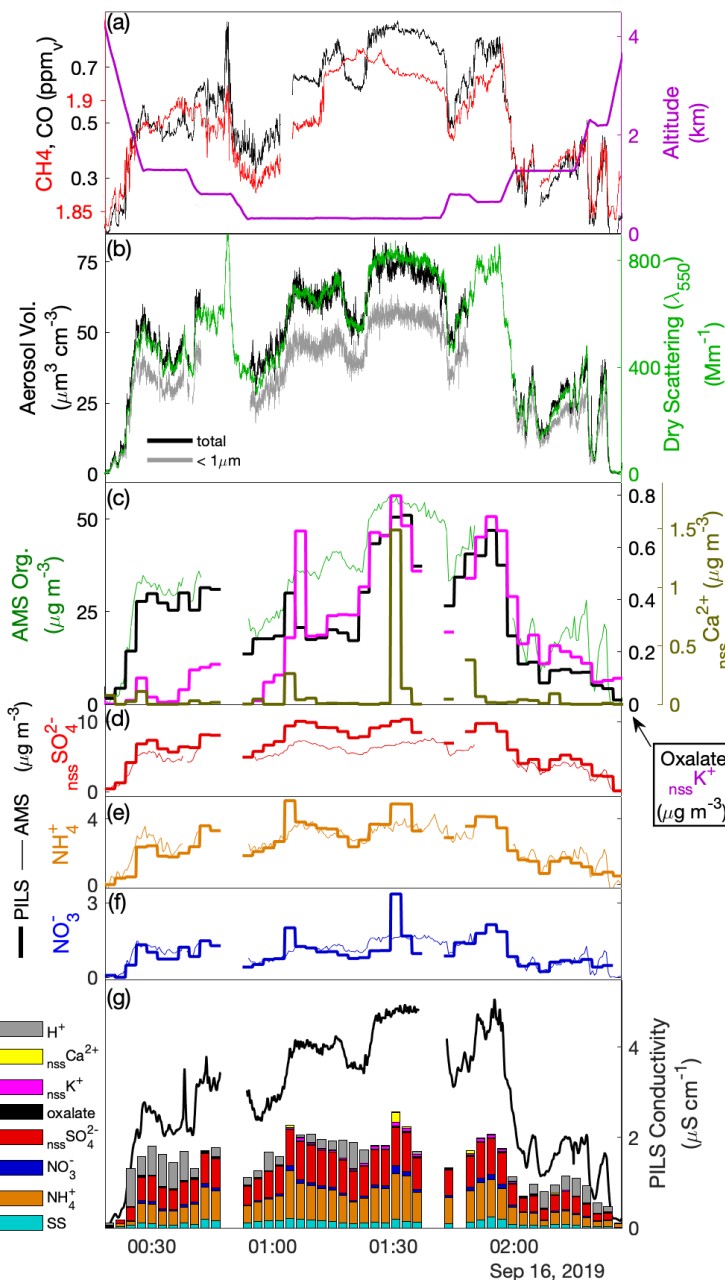

Figure 5: As Figure 3 but for the upwind cross-plume transect (TX) during Case III. Here, panels (a), (b) and (g) follow panels (a), (b) and (e) from Figure 3 but panels (c) and (d) have been replaced with: (c) comparison of AMS organic mass concentration with PILS biomass burning ion tracers (d-f) comparison of PILS and AMS nssSO$_4^{2-}$, NH$_4^+$, and NO$_3^-$,

respectively. The contributing species for the conductivity closure has been augmented.





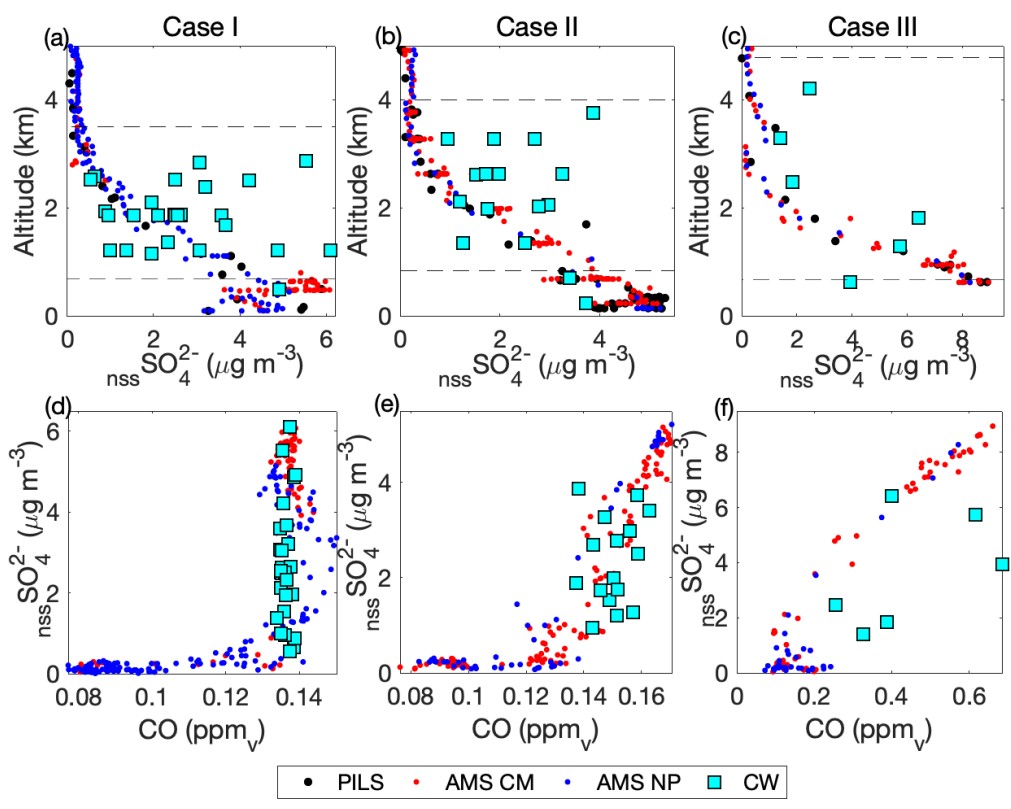

Figure 6: (a-c) Profiles showing the relationship between $_{nss}SO_4^{2-}$ and altitude for cloudy and clear air samples in Cases I-III, respectively. The colours, consistent with Fig. 2, represent AMS aerosol data in clear sections of the cloud module (AMS-CM, red), nearby profiles (AMS-NP, blue), and cloud water samples (cyan), and PILS data when available (black). The AMS data is scaled using a case specific factor (see text) to account for collection efficiency. For each case, an estimated cloud top and LCL are marked (dashed lines). (d-f) as (a-c), but showing the relationship between $_{nss}SO_4^{2-}$ and CO mixing ratio.





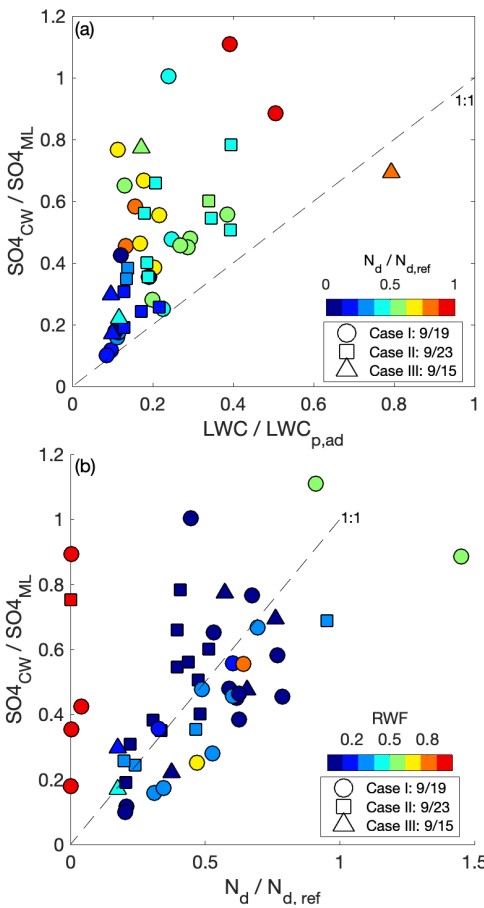

Figure 7: (a) Comparison amongst normalized CW $_{nss}SO_4^{2-}$, LWC and $N_d$. The normalization is with respect to an undilute parcel originating at cloud base and a reference droplet number concentration based on the 90[th] percentile. (b) The same comparison but amongst CW $_{nss}SO_4^{2-}$, $N_d$ and RWF.



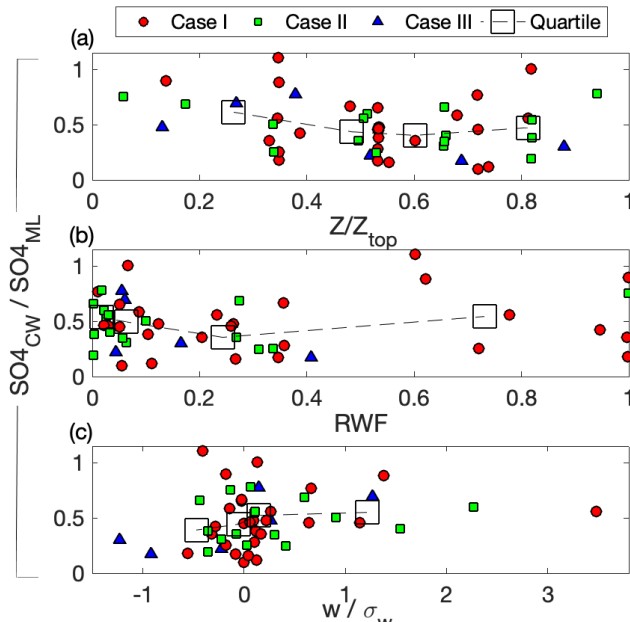

Figure 8: Variation of normalized CW $_{nss}SO_4^{2-}$ with (a) normalized altitude, (b) rain water fraction, and (c) vertical velocity (normalized by turbulent rms velocity, $\sigma_w$) across the three cases. In each comparison, the data are grouped into quartiles of the independent variable and mean values of the quartiles are shown.

l475

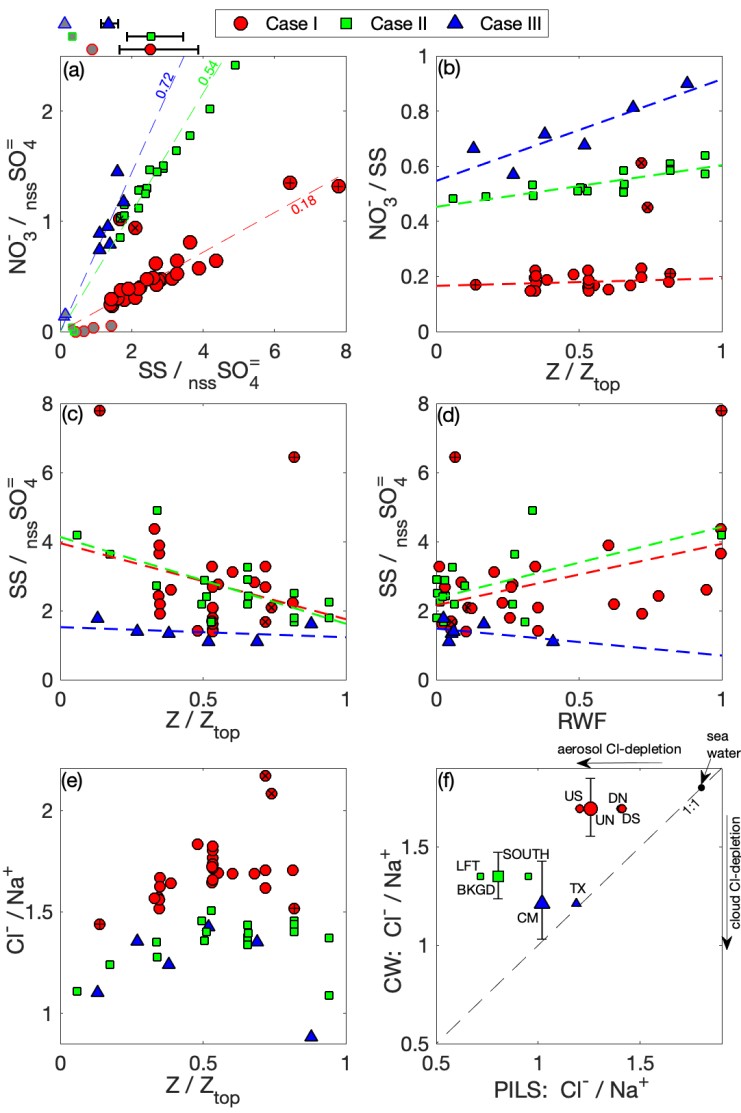

Figure 9: Relationships among SS, $NO_3^-$ and $_{nss}SO_4^{2-}$. (a) the CW mass ratios of SS to $_{nss}SO_4^{2-}$ are compared to mass ratios of $NO_3^-$ to $_{nss}SO_4^{2-}$ for each case, with relevant sub-cloud airmass PILS data included for contrast (grey fill). Above the panel, l480 the (geometric) mean SS:$_{nss}SO_4^{2-}$ mass ratio is shown (with 1σ variability) for CW for each case. Dash lines (with slope indicated) show case mean $NO_3^-$:SS mass; (b) CW $NO_3^-$:SS mas against normalized altitude; CW SS: $_{nss}SO_4^{2-}$ mass against (c) normalized altitude and (d) RWF; (e) CW $Cl^-$:$Na^+$ mass against normalized altitude; and, (f) sub-cloud aerosol (PILS) $Cl^-$:$Na^+$ mass ratio compared against equivalent CW $Cl^-$:$Na^+$ mass ratio. Case I with outlying ratios are highlighted: high-$NO_3^-$ pair (×), high-SS pair (+).



l485

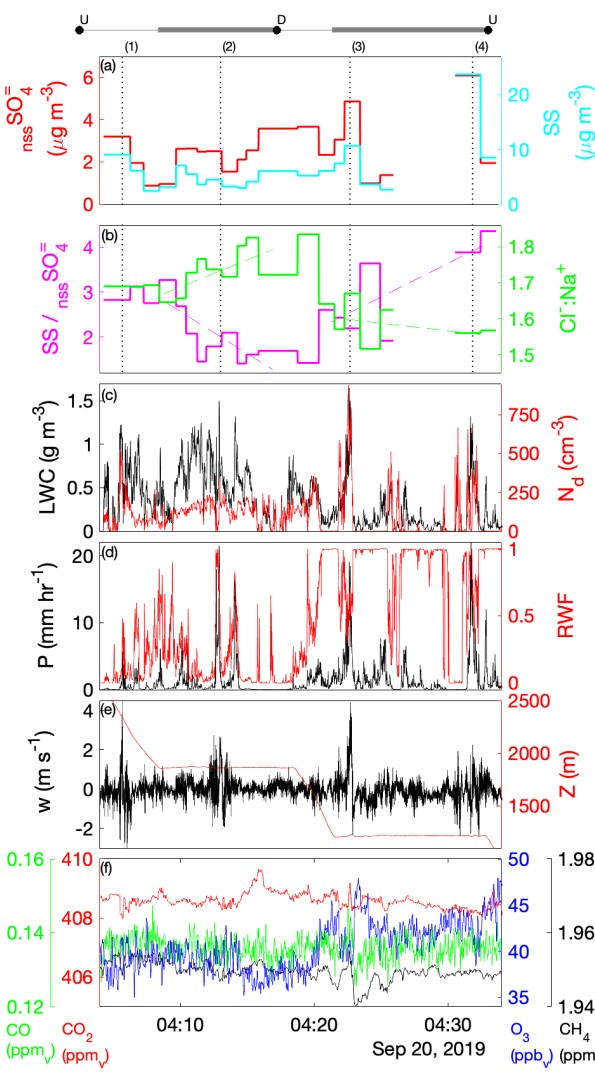

Figure 10: Time series of two mid-cloud levels flown during Case I, containing a high density of cloud water samples (20).
l490    The key at the top indicates when the aircraft was at the upwind (U) and downwind (D) extents of the cloud module and
        when on level legs (bold) – note that, unconventionally, descents between levels were not coordinated with the U and D
        turns. Panels show (a) CW $_{nss}SO_4^{2-}$ and SS mass concentration; (b) CW SS:$_{nss}SO_4^{2-}$ and Cl$^-$:Na$^+$ mass ratios; (c) LWC and $N_d$;
        (d) P and RWF; (e) vertical velocity (w) and aircraft altitude (Z); and, (f) gas tracers (CO, $CO_2$, $CH_4$ and $O_3$).  Also shown
        are 4 transects through convectively enhanced regions (dotted line in panels (a) and (b)), and the linear trends in the ratios
l495                                during the level leg sections (panel (b)).



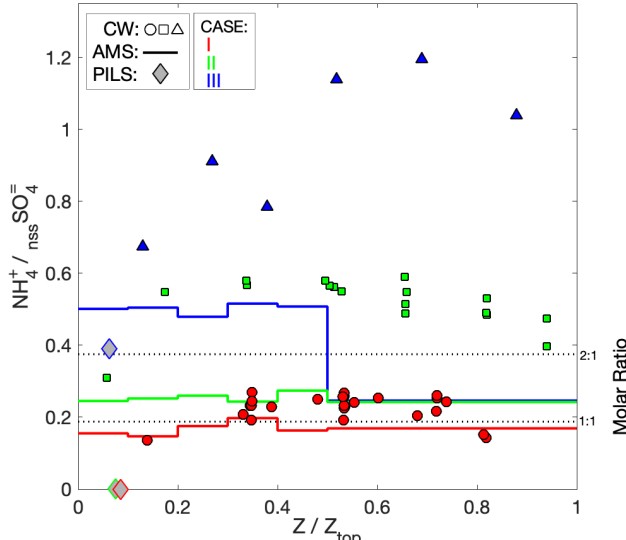

Figure 11: Profiles of CW $NH_4^+$ : $_{nss}SO_4^{2-}$ for each case. Also shown is (i) the mass ratio of mean concentrations of AMS $SO_4$ and AMS $NH_4$ computed in 5 altitude bins spanning the lower half of the cloud and 1 bin spanning the upper half of the cloud from adjacent clear air sampling; and (ii) the mass ratio calculated for PILS sub-cloud data.

1500



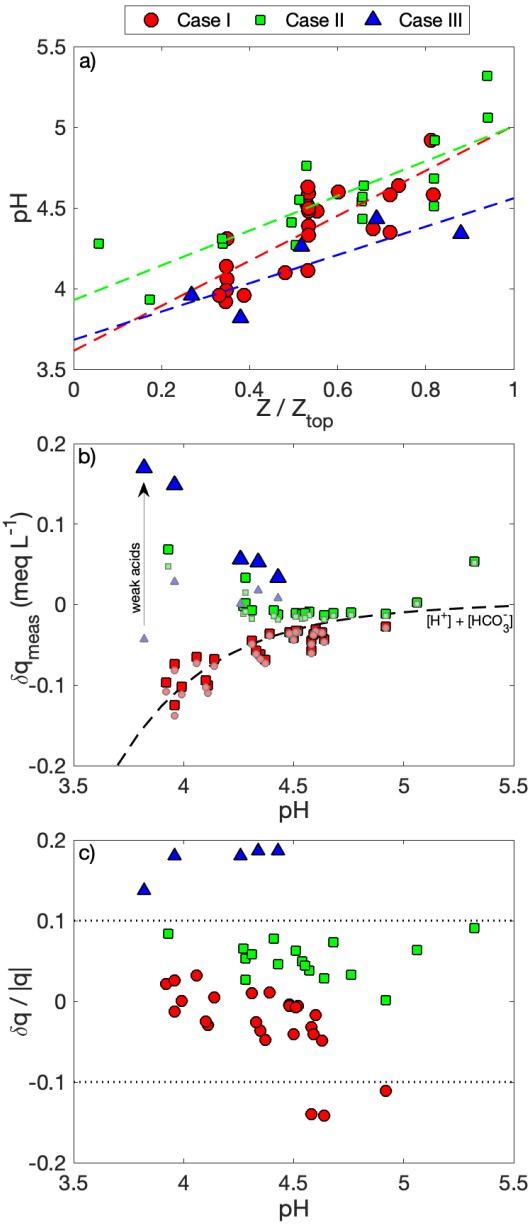

Figure 12: (a) Relationship of CW pH with normalized altitude; (b) pH-charge balance closure analysis: residual charges from measured ions are shown including (solid, large) and discounting (desaturated, small) weak acid dissociation, with the black dashed line indicating expected charge imbalance at each pH; and (c) residual charge from (b) normalized by the total measured charge. The region bounded by the dotted lines indicates a nominal +/- 10% zone that is likely indistinguishable from analytical biases.



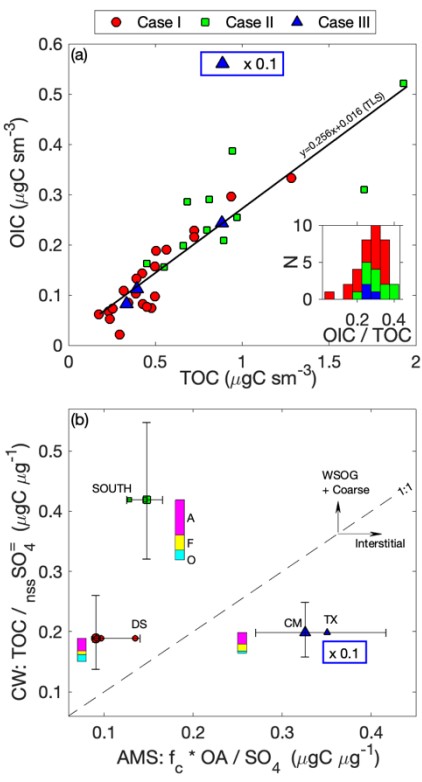

Figure 13: (a) Closure of the sum of the carbon contribution from each measured organic ion – the organic ion carbon (OIC) – against measured total organic carbon (TOC), with Case III scaled by a factor of 10 to aid readability. Inset is a histogram of the OIC:TOC ratio colored by case, and a total-least-squares (TLS) best fit line for the OIC-TOC closure is included. (b) Closure analysis for CW TOC and sub-cloud organic aerosol (OA) from the AMS. The CW data is normalized by CW $_{nss}SO_4^{2-}$

and OA data is scaled by an estimate of carbon mass fraction ($f_c$), and normalized by AMS $SO_4$. 'Airmass' means are included with a $1\sigma$ bar aligned with the 'airmass' most relevant to the cloud (as per Figure 9; here only outlying airmasses are labeled to aid readability). As panel (a), Case III is scaled by a factor of 10. The contributions to TOC arising from measured acetate (A), formate (F) and oxalate (O) are shown next to each case with the same normalization.





525

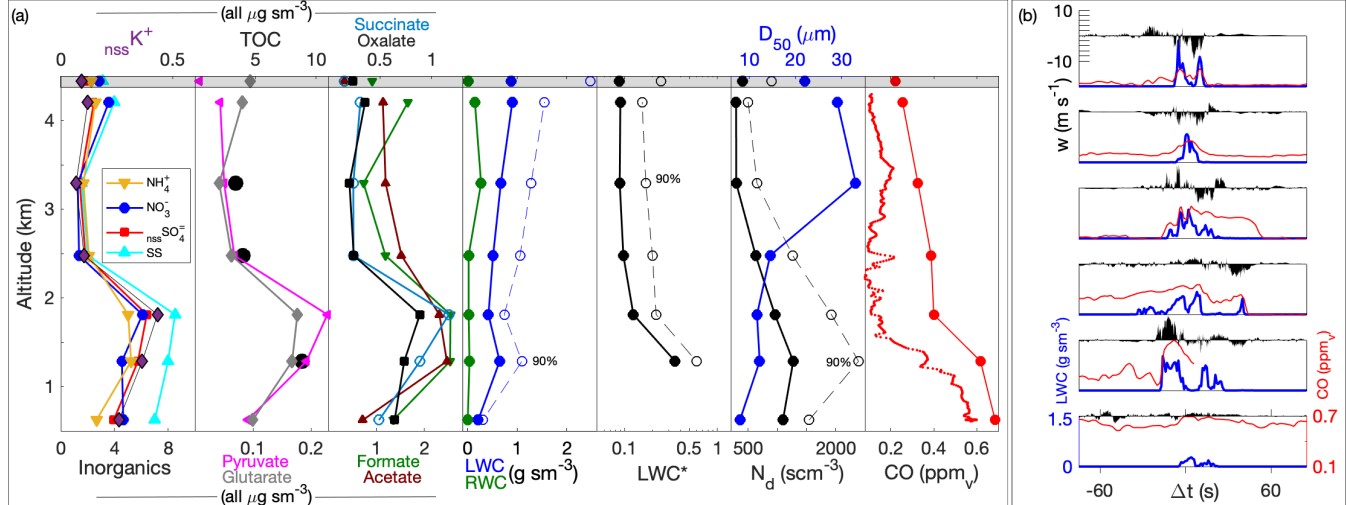

Figure 14: (a) Profiles of key CW species during the Case III cloud module and supporting microphysical (LWC, RWC, adiabatic ratio (LWC*), $N_d$, volume-based median diameter ($D_{50}$)) and gas tracer (CO) data. Also shown alongside CO is the CO profile in the nearby (cloud-free) environment. Included in the grey shaded region is the corresponding data from a cloud

530 top penetration near the upwind transect (yellow marker, Figure 2; see text). (b) Thumbnail time-series plots (+/- 75s from centerlines) of the six cloud penetrations corresponding to the CW samples showing LWC, CO and w, with all plots equally scaled.