# Peer review of "Measurement Report: Closure analysis of aerosol-cloud composition in tropical maritime warm convection"

_EGUsphere, 2022_

## Referee Comment (RC1)

Crosbie et al report on measurements of the chemical composition of cloud water and particles collected during the Clouds, Aerosols and Monsoon Processes – Philippines Experiment (CAMP2Ex). They focus on three cases that differed by their air mass type. The analysis is focused on various parameters and ratios. The paper is generally well written (with some minor exceptions; see below) and the methodology and analysis are explained in detail. However, the paper lacks a discussion that places the results into the context of previous studies. In addition, language is sometimes somewhat colloquial.

**Major comments**

1. The paper is very descriptive and the discussion is mostly limited to the comparison between the three cases and some attempts to explain their differences based on emissions and air mass type and history. With its current content, I recommend conversion to a Measurement Report.

2. If the authors prefer consideration as a Research Article, a discussion section should be added. The very detailed and nice introduction includes many useful references that highlight the complexity of aerosol cloud interactions and their importance in altering aerosol properties. The discussion of the presented results should be placed into context of such prior studies, either in a dedicated discussion section or as part of the conclusions. What are the main message and the main conclusions of this paper? What do we learn about the role of cloud processing that has not been known before?

3. The study is based on a very limited number of samples, which is inherent to the complexity, and difficulties of sampling in clouds. However, this sparsity of the samples and therefore the limitation of their interpretation should be also highlighted in the discussion section.

**Minor comments**

4. l. 26: The first sentence sounds awkward: (1) is composition a 'quantity' or 'property'? How do you 'quantify' composition? (2) Add 'chemical'.

5. l. 51: 'aerosols' in the parentheses seems redundant

6. l. 186: 'degree of freedom': Do you mean simply 'parameters' here or do you perform a statistical analysis that justifies this expression that is strictly used for statistical contexts?

7. l. 410, l. 415: 'AMS (PILS) composition' sounds colloquial. It is the 'particle composition measured by AMS (PILS)'

8. l. 470: This equation needs more explanation. Please also write it as an equation and not as in-line text.

9. l. 515: 'more variable' than what? As compared to the previous studies?

10. l. 684: 'SS subjected to aqueous production' implies that sea salt is formed by processes in the aqueous phase. Please clarify.

11. l. 808: Was Cruz et al's data collected at the same location?

12. l. 813: (1) If the acids are in the gas phase, they are not present in ionic form. Thus, it should read that 'formic and acetic acids reside in the gas phase'. (2) Is the assumption of complete evaporation justified? See for example, Liu et al., 2012, https://doi.org/10.1029/2012JD017912, Xiu et al., ACS Earth Space Chem. 2020, 4, 2, 157–167, https://doi.org/10.1021/acsearthspacechem.9b00210

**Technical comments**

13. l. 115: Define 'CAPE'

14. Several references are not included in the reference list: e.g., Cruz et al., 2019; Paciga et al., 2014

15. It would be very useful to provide a table with all abbreviations used, together with a short definition or equation, e.g. RWF, AEM, etc. It makes the reading very tedious to constantly having to flip back and forth to find these definitions.

16. At several places in the text, semicolons are used instead of commas. Semicolons should be only used to separate main clauses whereas commas can separate lists (e.g., l. 123-127, l. 152).

---

## Author Comment (AC1)

Crosbie et al report on measurements of the chemical composition of cloud water and particles collected during the Clouds, Aerosols and Monsoon Processes – Philippines Experiment (CAMP2Ex). They focus on three cases that differed by their air mass type. The analysis is focused on various parameters and ratios. The paper is generally well written (with some minor exceptions; see below) and the methodology and analysis are explained in detail. However, the paper lacks a discussion that places the results into the context of previous studies. In addition, language is sometimes somewhat colloquial.

Major comments

1. The paper is very descriptive and the discussion is mostly limited to the comparison between the three cases and some attempts to explain their differences based on emissions and air mass type and history. With its current content, I recommend conversion to a Measurement Report.

We agree that this would be an appropriate choice. In accordance with the guidance for preparation we have added "Measurement Report" to the start of the title.

2. If the authors prefer consideration as a Research Article, a discussion section should be added. The very detailed and nice introduction includes many useful references that highlight the complexity of aerosol cloud interactions and their importance in altering aerosol properties. The discussion of the presented results should be placed into context of such prior studies, either in a dedicated discussion section or as part of the conclusions. What are the main message and the main conclusions of this paper? What do we learn about the role of cloud processing that has not been known before?

We are in agreement with the reviewer's recommendation to prepare this as a Measurement Report and so broadening the scope to include these aspects seems to be more appropriate for a Research Article as the reviewer points out.

3. The study is based on a very limited number of samples, which is inherent to the complexity, and difficulties of sampling in clouds. However, this sparsity of the samples and therefore the limitation of their interpretation should be also highlighted in the discussion section.

It is not entirely clear what the reviewer is stating in this comment. The number of samples collected is not really driven by the complexity of the cloud scene, rather it is proportionate to time spent probing cloud systems with sufficient condensate loading for sample collection. The time spent conducting in situ sampling of clouds is a function of the demands of the mission, which, in this case, were directed to achieving a range of remote sensing and in situ sampling objectives.

The samples are not limited nor sparse, in the context of typical availability of cloud water samples collected by aircraft by the same or similar technology (e.g., the Mohnen rod). We do not think it is appropriate to describe the samples as sparse in any discussion about the three cases. In fact, we have described the opposite, for example, in Case 1 where there were 20 samples collected in rapid succession during two mid-cloud legs. A review of the literature on these measurements can help provide some baseline/context. For example, Benedict et al. (2014) VOCALS stratocumulus: 72 samples in 14 flights, Straub et al. (2007) DYCOMS2: 50 samples in 9 flights, MacDonald et al. (2018) 385 samples spread across 4 campaigns and 84 flights, Pratt et al. (2013) 2 samples, Hill et al. (2007) 20 samples

across 7 flights.  During CAMP2Ex we collected 330 samples across 19 flights, the cases studies presented here involved a subset of samples (51) collected during 3 flights.

Minor comments

4. l. 26: The first sentence sounds awkward: (1) is composition a 'quantity' or 'property'? How do you 'quantify' composition? (2) Add 'chemical'.

This has been revised, thank you.

5. l. 51: 'aerosols' in the parentheses seems redundant

Agreed, revised.

6. l. 186: 'degree of freedom': Do you mean simply 'parameters' here or do you perform a statistical analysis that justifies this expression that is strictly used for statistical contexts?

We have replaced "degrees of freedom" with "contributing parameters".

7. l. 410, l. 415: 'AMS (PILS) composition' sounds colloquial. It is the 'particle composition measured by AMS (PILS)'

Agreed, revised.

8. l. 470: This equation needs more explanation. Please also write it as an equation and not as in-line text.

The equation was included to avoid a text description.  Since the reviewer feels that it needs explanation, we have just provided a description rather than the equation.

9. l. 515: 'more variable' than what? As compared to the previous studies?

More variable, implying that the correspondence of these species to OA (e.g. as quantified by the ratio) was not a constant across the plume. Rather than add further description/clarification, we have gone ahead and truncated this sentence, removing the statement about "more variable".  The sentence now reads:

"Oxalate, $_{nss}K^+$, and $_{nss}Ca^{2+}$ were found to be enhanced in the plume (Figure 5c), consistent with other studies of biomass burning (Andreae, 1983; Yamasoe et al., 2000; Maudlin et al., 2015)."

10. l. 684: 'SS subjected to aqueous production' implies that sea salt is formed by processes in the aqueous phase. Please clarify.

This is probably a fair criticism of potential ambiguity in the statement as written.  We were certainly not intending to imply that sea salt was anything but a primary particle.  The intention was to convey that sea salt particles, in the form of aqueous droplets, are the site of aqueous phase reactions that result in production of acids. A simple rearrangement of the sentence removes the ambiguity:

"SS particles subjected to condensation, or aqueous production, of nitrate, sulfate and organic acids acids (e.g., Alexander et al., 2005; Chameides and Stelson, 1992; Sievering et al., 1992) tend to liberate…"

11. l. 808: Was Cruz et al's data collected at the same location?

No.  It uses ground-based measurements on land in the Philippines (near Manila), so it is within the region but not collocated with the aircraft measurements.

12. l. 813: (1) If the acids are in the gas phase, they are not present in ionic form. Thus, it should read that 'formic and acetic acids reside in the gas phase'.

It now reads:

"Although no supporting gaseous measurements were available, we make the assumption that acetate and formate reside as gaseous acetic and formic acid when outside cloud."

 (2) Is the assumption of complete evaporation justified? See for example, Liu et al., 2012, https://doi.org/10.1029/2012JD017912, Xiu et al., ACS Earth Space Chem. 2020, 4, 2, 157–167, https://doi.org/10.1021/acsearthspacechem.9b00210

Yes, we think it is justified for the purpose of this analysis. In the Liu et al. (2012) paper, the median partitioning ratio for both Atlanta and LA was of order 1%.  Furthermore, these urban cases have abundant $NH_3$ and the partitioning ratio for nitric acid shown in their cases nicely supports this.  We would expect the CAMP2Ex cases (specifically Case I and II) to have partitioning ratios that are lower still.

In terms of the budget of organics described in the following paragraph, the only impact on the reported fractions would be to (slightly) increase the residual fraction (i.e. that which we attribute to unmeasured water-soluble VOCs and organic material attributed to coarse particles), but there is no change to the conclusion that a significant residual (i.e. unclosed) fraction exists.

In our opinion, any attempt to make a more refined estimate of the partitioning ratio of formic and acetic is unjustified and potentially distracting given that it would be done without realistic and relevant case specific data inputs.

Technical comments

13. l. 115: Define 'CAPE'

Thank you for identifying this.  On reflection, this has been revised to "moisture and instability", reference to CAPE was superfluous.

14. Several references are not included in the reference list: e.g., Cruz et al., 2019; Paciga et al., 2014

Thank you for identifying this. Revised.

15. It would be very useful to provide a table with all abbreviations used, together with a short definition or equation, e.g. RWF, AEM, etc. It makes the reading very tedious to constantly having to flip back and forth to find these definitions.

Thank you for this suggestion. We have added an Appendix with this information.

16. At several places in the text, semicolons are used instead of commas. Semicolons should be only used to separate main clauses whereas commas can separate lists (e.g., l. 123-127, l. 152).

The use of the semi-colon as a so-called "super comma" in aiding readability for lists-within-lists (such as was flagged L123-127) is not unprecedented and aids readability. We are comfortable with the current usage of punctuation marks.

This manuscript represents an interesting dataset that covers a broad range of topics, and it does a nice job of demonstrating the complexities of conducting and analysing these types of studies. The breadth of the paper limits the focus that is put on any particular topic, and there is a distinct lack of discussion of the process of nucleation scavenging, despite this being fundamental to everything. Samples of water collected for chemical analysis are all classified as cloudwater (CW). A rainwater fraction (RWF) is used to identify the relative contributions from cloudwater versus rainwater, which I think is a nice approach. For a paper about clouds, I think there are too many thoughts about the discrepancies associated with the NH4+ measurement. Overall, the authors have a very nice dataset, including the information about organic components of the cloudwater, but I would like to see this long paper more focussed on a few issues that can be addressed well. For example, since there is disagreement on NH4+ among some of the measurements, eliminate NH4+ from the discussion and put it in another paper after you have completed your related laboratory investigations.

Thank you for the comprehensive review. We note the comment regarding nucleation scavenging and have provided responses and accompanying revisions alongside these individual comment bullets. We have considered the reviewer's suggestion that we remove $NH_4^+$ from the closure discussion but we do not feel that this is the appropriate course of action, mainly since $NH_4^+$ is an important component of both cloud water and aerosol particle composition. In summary we have three main concerns with following this suggestion:

a) In the "spirit" of composition closure, we want to report and discuss all measured components
b) $NH_4^+$ has implications for the discussion of neutralization of acidic species and gas-particle/droplet partitioning.
c) In our opinion, the aspects related to instrument closure between the PILS and AMS are important and should be reported. The research community may consider other avenues in addition to our reported hypotheses and proposed laboratory control experiments, and so we would like these findings to be included.

1. Introduction – There seems to be a lot of discussion about why observations don't work well, but then we are told that it is clear that they are needed. The discussion fails to acknowledge observations that have worked, mostly because of the nature of the clouds and well documented instrumentation.

This comment was a little challenging to interpret so we apologize if we have misunderstood the reviewer's intent. If so we will happily reconsider upon clarification.

We want to make clear the distinction between observational challenges that stem from (i) instrument limitations and/or sampling strategies, and (ii) the coupled nature of cloud-scale circulation, microphysics and chemical reactions that collectively affect the spatially and temporally varying abundance of species in the gas-phase, un-activated aerosol particles and cloud/rain drops.

The Introduction describes myriad cloud processes that result in production, loss and redistribution of particulate species. Our intent was to make sure that these processes were

suitably introduced because they affect the budgets of each species. A process that was given additional weight/discussion was aerosol removal by precipitation at the cloud system scale because, in the context of tropical warm convection, there is a tight coupling between precipitation, net latent heating and mesoscale vertical motion and this affects the manner in which this process could be observed. We argue, using examples, that complex cloudy trajectories in shallow convective environments pose a challenge for identifying regions of cloud processed air, such that measurements of abundances can offer insights into rates/tendencies. However, we also provide some examples of natural laboratory cloud cases where (local) cloud processing can be identified kinematically.

The focus on introducing these dynamical, microphysical and chemical processes does not imply that observations don't work well. Further, any success or shortfall in the ability to tease out process-level understanding from observations does not preclude the ability to use observations to accurately characterize cloud or aerosol composition and properties. In fact many of the observational studies listed do just that.

2. Lines 200-202 – Why were liquid drops "universally assumed", when you have 2D instrumentation you can use to avoid an assumption?

Yes, this is correct. We have cases (not described here) where cloud water was collected near and in the melting level and a more detailed calculation of total water content is needed. On reflection, this sentence is a distraction and we have removed it. Thank you for bringing it to our attention.

3. Line 218 – RWF is a ratio. Unless LWC includes RWC, the fraction should be RWF = RWC/(LWC+RWC). Please clarify.

LWC is not size limited. After carefully reviewing the description, we are convinced that there is no suggestion/implication that the integration of the drop volume distributions (used to estimate LWC) was supposed to be limited to a subset of sizes (e.g. to not include rain). The RWC is defined and the method of calculating RWF is defined and correct. In the event that this might need reinforcing to a reader, we have added "across all sizes" in describing the DSD integration for LWC.

Also, here it would be helpful to clearly state that all samples of water are referred to as CW with RWF used to indicate the relative proportion of rainwater in each.

Thank you this is a useful suggestion. We have added "All water samples are described as CW samples, whether or not they contain rain." into the earlier description of the CW samples (~L180).

It was not appropriate to add this to the description of the microphysical measurements (L201-229) because they are not "merged" to the CW samples and the stand-alone microphysical data and the RWF is used in some places at its native 1Hz time resolution (e.g., Fig. 10).

In Figure 6, identify samples that are RW in some way. For example, are the samples that appear to have been collected below cloud in Case II actually RW or is there some discrepancy in cloud base?

This is a useful addition. We have revised Figure 6 using symbols to identify high RWF CW samples that help explain samples collected outside of the "cloud" boundary. Note that we have used the LCL as a useful indicator of the cloud base region. We are sure the reviewer is aware that the envelope of a precipitating cloud, especially the description of its base, is ill-defined. After reviewing Figure 6, especially in light of adding the high RWF flag, we decided to consolidate/simplify the aerosol (cloud-free) profile data because separating the regions is not all that significant.

4. Section 2.5 - How were the blank samples, mentioned on page six, used? What were the detection limits?

They were used to assess any handling artifacts and material that could be leaching from the collector to the samples. We have added text to Section 2.5 to explain the blank subtraction.

5. Section 2.7 and Figure 1 – The implication here is that this represents one cloud. The very sharp and narrow downdraft just past 20 seconds suggests a possible outer edge of the larger cell. Does this time series represent two clouds in different stages?

First, let us repeat L266-268 from Section 2.6:

"In some instances, multiple CW samples were collected in a continuous block within the same contiguous cloudy region, while in other cases a single CW sample comprised partial volumes from several discrete cloud penetrations."

Cloud systems, even isolated convective elements/turrets contain multiple cores, and pulses that define their structure at any instance – they are not smooth. The boundary of a singular contiguous cloudy entity could be penetrated multiple times by an aircraft even on a level transect.

In Section 2.7 and the case shown in Figure 1, there is no implication or discussion of whether the core penetrations were conjoined (i.e., if a contiguous region is considered a necessary requirement) at the time of penetrations at the flight altitude, or any another altitude. Frankly, this type of distinction is not that useful and we have tried to avoid this type of identification or classification of what constitutes a singular "cloud" anywhere in the paper.

Larger convective cloud systems are potentially even more problematic, because they could comprise 10s if not 100s of convective elements as part of the circulation. These may be connected via detrained stratiform cloudy regions, precipitation or they may be initially separate, but evolve into the common cloudy mass. The definition of a singular cloud could take various forms.

So the answer to the reviewer's question is: equally yes, no, and it is not relevant to Section 2.7.

6.  Section 3 – For each case, it would be helpful to have a conceptual 2D cartoon showing the relative proximity of the clear-air profile to the cloud and the wind directions with heights. That is, something similar to Figure 6, but showing the proximity of the clear-air work to the sampled cloud.

This information is currently shown in Figure 2, in plan view.

7.  Line 416 – "PILS SS showed a marked reduction in US". Was that meant to be "DS", and reduction relative to what: in the DS intervals, SS:nnSo4= appears to be higher?

No, the description is correct as is and reflects the data in Table 2. The concentration of sea salt for the US airmass was lower than the three other airmasses which were relatively similar.  We have rearranged the sentence to try and remove this ambiguity.  It now reads:

"PILS SS showed a marked reduction in US, compared to the relatively consistent concentrations observed in the other airmasses, and this reduction was reflected (qualitatively) by proxies for coarse particles (V-LAS$_{>1\mu m}$, V-APS, V-FCDP, N$_{APS}$, N$_{FCDP}$)."

8.  Lines 472-473 – The statement "A large number of airmass properties that characterize sub-micrometer aerosol" seems a bit odd.  Should it be reversed: i.e., "A large number of sub-micrometer aerosol properties that help to characterize the airmass"?

Yes thank you for highlighting this. We have revised this phrase to read:

"A large number of observations that characterize the properties of sub-micrometer aerosol…"

9.  Lines 506-507 – With an hour and a half between the stacked legs, should you expect consistency?

We did not have particular expectations for the vertical structure and how it may evolve in time and vary across the ~250 km between the two series of stacked legs. In this sentence we are providing a supporting description for the previous statement that broadly described concentration variability and we are essentially noting here that not only is there cross track heterogeneity but also that there is complex and variable vertical structure.

While considering this comment, we felt that the previous sentence (that this supports) was awkward and so we have revised it to make it clearer:

"A time series of a crosswind transect across the biomass burning plume (Figure 5) illustrates the structure of the plume and shows concentrations that were highly variable both in the vertical and horizontal."

10.  Lines 520-522 – This does not appear to be true, as increases in nitrate coincide with

the spikes in calcium, which you note later in this paragraph in connection with the possible reactions of HNO3 with insoluble (or less soluble) calcium-containing soil molecules.

Thank you for raising this. We have adjusted the statement to better reflect our intended meaning, which was mainly the comparison with the species in Fig 5c.  The sentence now reads:

"…$_{nss}Ca^{2+}$ exhibited periodic high anomalies (e.g., 0106 UTC, 0130 UTC) but exhibited neither correlation with the other data shown in Figure 5c nor covariability with microphysical coarse-mode proxies (e.g., V-APS, V-FCDP) that could confirm similar fine-scale structure in crustal material."

Related - lines 525-529 – Why would insoluble components be responsible for the spikes if they're insoluble?

First, we felt that it was important to discuss the structure of nitrate especially since the concentrations measured by the PILS departed from the AMS in a few localized regions that also coincided with anomalous spikes in calcium. Dust/soil particles may be expected to contribute to calcium and nitrate mass, while also not contributing to nitrate measured by the AMS (insensitivity to refractory mass, and potentially because of size).

However, the spatial structure of the calcium (+ nitrate) anomalies is not supported by microphysical data that could be used as proxies for dust/soil particle mass, namely the integrated volume from the APS and FCDP.

In summary, it was important to report the linkage between nitrate and calcium anomalies, but it was also important to report that there was no further support/agreement for the spatial structure of these anomalies.

Returning to the reviewer's question: in L527-529 we offer a hypothesis to explain the behavior based on the fact that we know that it is possible for insoluble material to accumulate based on the Wonaschutz study.  Solubility may be affected by a number of factors (e.g. temperature, acidity, concentration of other ions in solution), so inherent in the hypothesis is that if crustal deposits are released into the PILS sample stream some fraction may be soluble enough to contribute to enhancements in subsequent offline measurements of calcium and nitrate.

While we want to ensure there is no ambiguity, we are also keen to avoid placing too much emphasis on this explanation with further details because it is merely a hypothesis for a structural disagreement.  We have replaced "insoluble deposits" with "low-solubility crustal deposits".

11.    Lines 584-586 – Why is sulphate a useful candidate for closure, since in-cloud production of sulphate is a dominant factor contributing to global sulphate?  Also, if you are to undertake a closure analysis of sulphate, you must consider the fraction of sulphate scavenged by nucleation.  It would seem from the many measurements you mention,

including size resolved sulphate from the AMS and droplet distributions, that you have the tools to include nucleation scavenging in a reasonable manner.

Note: the AMS is not run in P-TOF mode because of the desire to get fast time response data on the aircraft, so we do not have sulfate mass size distributions.  Volume size distributions can be estimated from the LAS measurements assuming spherical particles.

The question of what mass fraction we might expect to activate into droplets at cloud base is recurring through the remaining section of this review, and so we take the time here to perform calculations that provide addition information in support of our methods.

We have performed a series of calculations in support of the expected scavenged fraction (by nucleation) in an idealized updraft near cloud base.  The first approach is to use an upper bound of the measurements of cloud droplet number concentration (we use the 90[th] percentile as a statistical maximum) as an estimate of the activation conditions in an undilute parcel.  Note that this estimate is not just during instances of cloud water collection, we survey all the cloudy data to attempt to find a best estimate of conditions that may closely resemble an undilute parcel.

This droplet number concentration then is compared to a cumulative distribution of particle number (i.e., total number concentration greater than threshold size as a function of threshold size) to provide an estimate of an activation diameter.  This estimate makes the assumption that when a peak supersaturation occurs the particles whose dry diameter is larger than the activation diameter will continue growing into large droplets because the peak supersaturation exceeds their critical supersaturation, while the particles whose dry diameter is smaller than the activation diameter do not ever achieve their critical supersaturation thereby remaining unactivated. The estimate does not account for joint distributions of hygroscopicity and size and essentially treats particle size as the only discriminator for activation.  Using this method we find activation diameters for the respective cases to be 126 nm, < 94 nm and 172 nm respectively. For Case 2, we could review other available particle size information to attempt to try and determine how much lower the activation diameter is likely to be, but, on a mass basis, it is irrelevant.

Now we consider these activation diameters with respect to the volume size distribution and make an estimate of the fraction of particle volume (hence, an estimate of mass) that is incorporated into activated droplets. Recognizing that the sulfate mass size distribution may not conform to the same distribution shape as the entire volume distribution inferred from the LAS measurements, we truncate the estimate at 1 µm.  This results in mass fractions (i.e. translating into scavenged fractions at nucleation) for the three cases of 0.97, 0.99, 0.98.  Note that in Case 3 (biomass burning) the volume size distribution is shifted quite a bit to larger sizes compared to the other cases.  For information, the volume modes of the three cases are 282 nm, 251 nm, 446 nm.  Instead of using 1 µm we can assess the sensitivity of this assumption and assume that all the sulfate is contained below 500 nm, but this makes little change in the estimated mass fraction.

A second approach was taken that avoided any reliance on an estimation of the cloud droplet number.  We seek an estimate of the lowest peak supersaturation that would be physically reasonable for these convective cloud scenarios.  The use of a lower bound

allows for an upper bound on the activation diameter, thus providing a conservative lower bound on the scavenged fraction at/near nucleation.  Consider 0.5% supersaturation, given that these are energetic clouds.  We use the "kappa" form of the Kohler curve with an assumed kappa=0.1.  This is likely a low (and therefore conservative) estimate for Cases 1 and 2, but ought to be in line with expectations for Case 3. The critical diameter for 0.5% supersaturation is 164 nm.  If we apply this threshold to the volume distributions, the estimated scavenged fractions are 0.95, 0.92, 0.99.

So we can conclude that even with quite conservative assumptions for a lower bound we could expect the scavenged fraction to be high at inception into cloud.  Let us now consider published literature.

Here is the abstract from Jensen and Charlson (1984) with the most relevant section highlighted:

*"The activation process near cloud base for submicrometer, soluble atmospheric aerosols is described and special attention is focused on the fraction, $\varepsilon$, of the total aerosol mass concentration that form cloud droplets. The variation of $\varepsilon$ with updraft speed for various aerosol size distributions is calculated by means of an adiabatic, one-dimensional Lagrangian cloud model. Calculations are made for aerosol particles consisting of pure ammonium bisulphate, (NH,)HSO$_4$, and the results show that for a clean continental **background aerosol population. $\varepsilon$ should be very near unity for all updraft speeds**. For an average urban aerosol population, **$\varepsilon$ is close to unity for convective clouds**. but for stratiform cases $\varepsilon$ decreases rapidly as the updraft velocity is lowered."*

Reviewing Jensen and Charlson's calculation for activated fraction in their polluted scenario for 1 m/s updraft yields >90% (their Figure 5).  Their polluted stratiform regime is not applicable to our cases, but this is the only scenario where a significant reduction in activated mass fraction ought to be expected.  The mass loading of their polluted cases is markedly higher than our Cases 1 and 2.

It should be clarified here that their definition of nucleation scavenging is the same as ours and we specifically establish this at the outset of our Introduction when we introduce the term nucleation scavenging. In a later comment (comment 20) the reviewer references Jensen and Charlson (1984) in connection with the expectation that sulfate not be fully incorporated into cloud water, but we cannot make this connection.  Let us consider some other references:

Leaitch et al. (1986) provide some data that we can contrast with our cases.  They report a 73% average scavenging ratio (evidently synonymous with scavenged fraction) for their winter and summer data using CW versus ground level aerosol sulfate, but they state: "the result presented here represents a complete cloud scavenging ratio" and by that they mean that it may include sulfate production but, crucially, it is not a measure of nucleation scavenging per the Jenson and Charlson (1984) calculation; it is the manifestation of mixtures aggregated across the sample, which they attribute to ~57% on a drop number basis. Mixtures may translate differently between drop number and solute mass depending on the nature of mixing (i.e., the spectrum of homogeneous to extreme inhomogeneous), which we have discussed previously, and size dependent solute concentration.  We have purposefully avoided describing the ratio of cloud water sulfate to sub-cloud aerosol sulfate (e.g. the ratio that we show in Figure 7) as a "scavenged fraction" or "scavenging ratio"

because that discounts other processes such as precipitation (which admittedly may not affect/be a leading order term in the study of Leaitch et al. (1986)), as well as dilution which is ubiquitous and sulfate formation which may or may not be a leading order term. Hegg et al., (1984), also report a "total scavenging coefficient" defined equivalently to our ratio in Figure 7. Their mean value (0.7±0.2) is similar to Leaitch et al. (1986) but here is how they define the scavenging coefficient:

*"The ratio of the mass of $SO_4^{2-}$ contained in the cloud water in a unit volume of air to the mass of $SO_4^{2-}$ in a unit volume of non-cloudy air was taken to be the total scavenging coefficient for $SO_4^{2-}$ ($F_s$)."*

They then note that this includes contributions from nucleation scavenging and sulfate production, which they then attempt to separate. Their analysis discounts any influence from entrainment, the resultant dilution of a cloudy parcel, and any adjustment to the scavenged fraction caused by evaporation affecting the cloud water sulfate during the time between cloud inception and sampling. Please note that we intentionally do not cite Leaitch et al. (1986) or Hegg et al. (1984) when introducing the concept of nucleation scavenging exactly because of this ambiguity.

When introducing the concept of nucleation scavenging Hegg et al. (1984) note:

*"…0.1 μm dia. particles (and thus virtually all of the $SO_4^{2-}$ mass in the air) will be activated at a supersaturation of 0.5%. Such supersaturations are achieved in clouds containing modest updrafts of 0.5-1.0 ms$^{-1}$"*

which is in close alignment with the findings of Jensen and Charlson (1984) and indeed of our calculations above. However, when describing the cloud types studied, they note:

*"With regard to the cumulus clouds, recent research has revealed that such isolated, moderate cumulus undergo little lateral entrainment and thus the air entraining into the cloud at its base is representative of that in which the cloud hydrometeors form, at least in the lower portions of such clouds where we sampled"*

This viewpoint on entrainment is perhaps outdated in light of considerable research over the past 30+ years. A more recent review by de Rooy et al. (2013) on this topic offers a comprehensive overview based on both observations and models.

This may offer some explanation as to the choices and assumptions made in Hegg et al. (1984) and Leaitch et al. (1986), but we have been explicit in separating and stating the importance of these processes in our Introduction (e.g. see lines 91-99). We are not suggesting that it is not possible (in fact it is probable) to observe scavenged fractions that are ~0.7 (and lower) at various locations in all three cloud systems of our study – but this is exactly the purpose of the analysis covered in Section 4.2. We are just not treating: (i) the ratio of in-cloud sulfate to sub-cloud sulfate, (ii) scavenged fraction, and (iii) nucleation scavenging as the same thing.

A further study worth mentioning here is van Pinxteren et al. (2016), which we reference when introducing natural laboratory scenarios. In this case, the study is a mountaintop experiment with orographic cloud. Here, they have the advantage of measuring both interstitial and droplet species concentrations, while also leveraging more constrained parcel trajectories/streamlines making their estimation of scavenging efficiency (scavenged fraction) more suitably aligned with conditions at cloud inception and therefore useful for

treating scavenged fraction as synonymous with nucleation scavenging.  Here are two relevant excerpts:

*"Mean in-cloud __SEs for sulfate are usually > = 0.9__ except for FCE11.2 and FCE13.3, where substantial fractions (21–44 %, depending on data used) of in-cloud sulfate reside in interstitial particles. During these events particle activation curves obtained from comparing measured particle number size distributions upwind and in-cloud were comparably shallow and the critical activation diameter was larger than during other events (Fig. S7), consistent with larger fractions of submicron sulfate not being activated to cloud droplets due to cloud microphysical conditions."*

*"In conclusion, the comparison of upwind and in-cloud scavenging efficiencies reveals that (i) nucleation scavenging typically removed > 80 %, often close to 100% of soluble material from the particle phase upon cloud formation,…"*

The advantage of the definition used in van Pinxteren et al. (2016) is that they leverage direct measurements of the interstitial mass such that they can constrain the total sulfate locally (rather than the total sulfate below cloud base, used in Hegg, Leaitch and our study), which makes treatment of dilution simpler.  However, they can only treat scavenged fraction (or scavenging efficiency) as meaning the same as nucleation scavenging because of the specificity of their orographic cloud sampling location.

If the reviewer can provide us with any examples that actually show measurements of nucleation scavenging (when considered in the same manner as Jensen and Charlson 1984, or this work; i.e., the scavenged fraction shortly after incorporation into cloud) that suggest sulfate could take on values as low as say 0.5 (as indicated in comment 18) then we would be very happy to review these cases and contrast them with the cloud cases we have studied.

Returning to the first part of the reviewer's comment in which they posed the question: Why is sulphate a useful candidate for closure, since in-cloud production of sulphate is a dominant factor contributing to global sulphate?

Perhaps the first comment to make is that none of the measured species are truly ideal candidates for closure.  Such an ideal candidate would be a non-volatile, primary aerosol species with no secondary sources that is externally mixed from all other species and found in particles that are large enough to assume near complete nucleation scavenging at cloud inception at all (reasonable) supersaturations, but at the same time small enough to be fully captured by the size limitations of the aircraft instrumentation.

The mass concentration of this idealized candidate, measured in CW, should then reveal four processes when compared with its concentration in out-of-cloud source airmasses: (i) the dilution from mixing of environmental air, (ii) the complete evaporation of a subset of droplets (e.g., following environmental mixing) resulting in a reduction in the scavenged fraction (i.e., from near 100% at inception), (iii) a recovery in the scavenged fraction during a recirculation event and following evaporative loss that, say, re-establishes a state of supersaturation, (iv) loss and redistribution from precipitation.

Unfortunately, we do not have any tracers that exactly meet all the requirements for the ideal candidate. We have two choices: either we consider each measured species/group in turn and discuss the four processes above in concert with deviations that make it non-ideal (e.g., semi-volatile, chemical production in cloud), or we take a candidate species that meets most of the ideal requirements, evaluate it as a "pseudo-ideal" tracer, and then evaluate the other species in a relative sense.

We have elected to use the latter approach because it allows us to (a) assess species specific deviations from the characteristics of the "pseudo-ideal" tracer and (b) eliminate factors that are common to all species, including some contributions to uncertainty like the assignment of LWC to a CW sample.

For sulfate, the major distinction from the ideal tracer is that there are secondary sources from in-cloud production. The second distinction, which hopefully has reached some level of resolution following the discussion above, is that potentially we cannot assume that all sulfate activates into droplets at inception. We need to stress however that (ii) and (iii) listed above can continually modulate the scavenged fraction under convective circulations, while (i) reduces the total sulfate (i.e. CW and interstitial) in a cloudy parcel below the total sulfate found in the sub-cloud region, assuming that the sulfate in the surrounding environment is lower aloft.

Sulfate production could be a leading order term if these cases were closer to pollution sources. The three cases involve airmasses that have been aged for several days from the pollution source that is responsible for their aerosol enhancement. Since in-cloud production of sulfate is rapid (compared to this aging time), we can reasonably assume that most $SO_2$ from the pollution sources has been consumed, leaving only precursor emissions associated with the regional background to resupply $SO_2$ for new production. It is hard to make the case that these rates would generate a strong perturbation given that these are already moderately (Cases 1 and 2) and strongly (Case 3) polluted cases.

We have made significant changes to Section 4.2 in an attempt to clear up some of the issues raised by the reviewer. The main changes are:

1) We define a budget for a cloudy parcel
2) We define the scavenged fraction
3) We recast the budget in the framework of the normalized quantities later discussed in Fig 7 and 8
4) We add a description summarizing the use of sulfate as a pseudo-ideal tracer following the rationale discussed above
5) We add a note about the calculations of activation diameter and the scavenged fraction at nucleation
6) We have added some more analysis of the comparison with droplet number including the explained variance
7) We note the expectation that production rates, accumulated over a typical cloudy trajectory, are not likely to create a significant perturbation to existing abundances given the age of the airmasses and the limited resupply of precursors.

12.    Line 598 – Your statement about the implication of "positive curvature" is interesting, but I would like to see a little more discussion here.  It seems reasonable that the wide variation of SO4= at a near constant CO of about 0.135 ppm may be indicative of precipitation scavenging (Case I), but there can be influences affecting SO4= as well as CO.  In making your statements in this section, your discussion should consider major losses and sources of CO and SO4= in order to rule out other explanations for the behaviour of the clear-air curves in Cases II and III.

CO and $CH_4$ are used as inert tracers in this study. The lifetime of both gases far exceed the timescales considered and both are negligibly soluble.  In cloud, there is no source of CO and $CH_4$ thereby making comparison of aerosol species to CO a valuable method of isolating convective transport of source airmasses (i.e., sub-cloud air together with entrainment of surrounding environmental air).  Given that these are aged airmasses, a foundation assumption can be made that the CO concentration represents a mixing fraction of polluted boundary layer air with a free-tropospheric background.

Convective mixing exerts the same influence on aerosol mass as the CO tracer.  Chemical production results in an enhancement of sulfate over that predicted by convective mixing manifesting as negative curvature. Precipitation leads to a deficit in sulfate over that predicted by convective mixing manifesting as positive curvature. We are not aware of any process that we have missed.

Note that we have made considerable changes to Section 4.2 in light of comment 11 (and companion comments below)

13.    Lines 709-611 – I think this statement belongs in section 2.7.

A good suggestion, thank you.

14.    Lines 613-622 – It is here that you first distinguish between CW samples collected in cloud and CW samples collected in precipitation in otherwise clear air ("unsaturated precipitation" as you call it, which is might better stated as precipitation in unsaturated air.)  RW is subject to some additional processes than CW.  As I mention above, the distinction should be shown in Figure 6.

We have amended per your suggestion.  Comments regarding Figure 6 were addressed.

15.    Lines 621-622 – I would appreciate a more detailed explanation of the statement that "The singular Case III data point… is perhaps indicative … that … a reference parcel with 100% scavenged fraction is unrealistic, even under undilute conditions, given the competition for water vapor in the highly particle-loaded smoke plume."  What is a reference parcel with 100% scavenged fraction?   What does competition for water vapour have to do with the reference parcel?

The response to comment 11 should provide some assistance and the revisions to Section 4.2 should make this clearer in the paper. The discussion about that singular data point was

actually removed during the revision of Section 4.2. and added into the discussion about the organic to sulfate ratio in Section 4.3.4

If the peak supersaturation in Case 3 were more strongly suppressed below expectations for convective clouds and actually resulted in a meaningful reduction in the mass fraction scavenged at nucleation, then the reason for that would be intense competition for water vapor.

16. Lines 624-635 – Are the quartiles in SO4=CW/SO4=ML different across Z/Ztop, across RWF and across w/sigmaw after taking into account the uncertainties in the measurements and sampling constraints (e.g., how representative is your SO4=ML of SO4= at cloud base?)

The reason for the detailed analysis of the sub-cloud airmasses was to determine the most appropriate estimate of the below cloud base environment. Any uncertainty would not affect the quartiles in a meaningful way, because the use of the normalization was just to allow the three cases to be evaluated together.

Note that of the 19 CAMP2Ex flights, these three cases were selected because they offered the best potential for minimizing this type of uncertainty (i.e. there was sufficient time sampling at low altitude below and around the cloud system, and the in-cloud measurements were close to where the sub-cloud measurements took place). Aircraft measurements have inherent bias in the sampling but we are not going to attempt to quantify this as an uncertainty.

17. Line 635 – By "scavenged" here, do you mean nucleation scavenged?

No, please see response to Comment 11.

18. Line 641-642 – There are at least two reasons why I don't think you can make this statement about nssSO4=:

The statement we have made in these lines is only about the measurement of aerosol sea salt and aerosol sulfate. We have added the word "aerosol" (i.e., total aerosol abundance) just for the avoidance of any doubt. The two reasons below are legitimate potential contributing factors for assessing the difference in sulfate:seasalt between aerosol and cloud, but neither of these factors influence the statement being made; that is, that we do not capture the entirety of the sea salt size distribution with the PILS.

a. You don't appear have sufficient information to help assess the importance of sulphate production in cloud.

b. You haven't assessed nucleation scavenging at cloud base. If only 50% of the SO4= is scavenged, some aspects of the assessments arrived at from Figures 6-8 could be quite different.

19.    Line 642-643 – Four microns seems to be around the cut off for sampling coarse particles from many aircraft, but the statement about "grossly underpredicted" sea salt particles should be either toned down or better supported.  You have measurements to give you information on the profiles of coarse particles that can be used to help assess the potential contribution from larger sea salt particles to your cloudwater samples.

We have removed "grossly" to tone down the statement, as suggested.

20.    Line 649 – "as anticipated" – We expect sea salt particles to be entirely incorporated into cloud during nucleation because of their generally larger sizes.  We do not expect SO4= particles to always be entirely incorporated (Jensen and Charlson, Tellus, 1984; Leaitch et al., Tellus, 1986).  For that reason, it is possible that your SS/SO4= can be higher in the CW than in the PILS samples.  You bring up nucleation scavenging occasionally, but you do not try to address it in anyway with your measurements, despite it being a major process impacting your discussions.

This comment appears to be continuing the theme of Comment 11 – please refer to the response to Comment 11 and the revisions to Section 4.2.  We consider these references in that response.

Let us first start by writing down three contributing reasons for why SS:sulfate could vary between PILS and CW.

  a)  SS measured by PILS does not fully capture the entire size distribution
  b)  SS is larger than sulfate and therefore this could result in difference in the scavenged fraction between the two components. Even if both components were entirely incorporated into cloud in an updraft near cloud base, the resulting complex trajectory of a cloudy parcel between inception and sampling could promote this difference (e.g. through subsaturated conditions in downdrafts and repeated recirculation
  c)  Sulfate is produced in cloud

All of these contributing factors are anticipated, and evidently a simplified form of (b) only considering cloud base is anticipated by the reviewer.  We are not sure if this comment is just a concern about the inclusion of "as anticipated" or a broader concern about the relative importance of the three factors above.  We have removed "as anticipated" because it is not needed in the sentence.

Let us now consider the comment in relation to the broader question of the relative importance of the three factors above, although please recognize that this sentence does not make claim about the relative importance.

If (a) was not a factor – as perhaps the reviewer is indicating – then we would have selected sea salt as the best approximation of the ideal tracer because items (b) and (c) are not a factor. However, if we attempt to compare CW SS to PILS SS, we find that CW SS exceeds PILS SS in a large number of samples, which presumably is evidence to the importance of

(a), since there are no sources for sea salt in cloud apart from that which is lofted upwards from below.

As we noted in response to comment 11: we could have treated each species in turn and evaluated the individual mass between cloudy and clear airmasses; however, we did not choose to do that. Instead, we chose to anchor one species (sulfate) then conduct the remaining comparisons on a relative basis.

The reviewer should note that immediately after the sentence that sparked this comment, we state the following (L650-652):

"The variability seen in CW SS:$_{nss}$SO$_4^{2-}$ can generally be attributed to some combination of the effects of $_{nss}$SO$_4^{2-}$ in-cloud production, differential scavenging and precipitation loss mechanisms and, potentially, highly localized variability in sea spray."

We have also added the following text to further reinforce the mechanism described as differential scavenging:

"Differential changes in scavenged fraction could manifest as a result of circulations within the convective cloud system that promote regions of sub- and super-saturation along a cloudy parcel trajectory. Similar behavior was demonstrated by Jensen and Nugent (2017), who discussed the condensational growth of large sea salt particles in downdrafts."

Thus, following inception in cloud, the scavenged fraction between seasalt and sulfate may diverge because of differences in condensation/evaporation under time-varying super/sub-saturation along a cloudy trajectory, the ratio may also vary as a result of differential sources (e.g. sulfate production) and differential sinks (e.g. propensity to form precipitation or be collected by falling drops). This could be viewed in the context of the normalized budget equation in Eq 3. (see changes to Section 4.2)

21.  Line 669 – "susceptible to washout" should be referenced, e.g., Seinfeld and Pandis.

We specifically defined all these terms in the Introduction so as to avoid (a) any doubt about the terminology (b) so they are defined and referenced. Here is the relevant text:

" Incorporation of aerosol mass into cloud water through activation of cloud condensation nuclei (CCN) – known as "nucleation scavenging" – (Jensen and Charlson, 1984) and the contribution from impaction/diffusional uptake of interstitial particles by cloud droplets (Flossman et al., 1985), leads to subsequent removal, subject to physical conversion of the cloud condensate to surface precipitation – known as "rainout" (Radke et al., 1980; Flossman et al., 1985). **Alternatively, cloud and rain drops can release the scavenged material upon evaporation (Mitra et al., 1992; Wang et al., 2020) or collect additional material in subsaturated environments (e.g., below cloud base); however, this mechanism – known as "washout" – has reportedly variable importance (Bae et al., 2012; Aikawa and Hiraki, 2009; Andronache, 2003; Croft et al., 2009) depending on both aerosol particle size and rain rate (Andronache, 2003).**"

22.  Lines 669-670 – "CW SS in unsaturated environments". Are you proposing to change

terminology: i.e., rain becomes cloudwater in unsaturated environments? This complicates reading.

We understand the complication, thank you for raising this point. We have revised the statement to read "…progressive enrichment of SS in rainwater below cloud base…"

23.   Line 676 – Just the 2nd leg, not "The lower (2nd) leg". It is not lower than 3 or 4.

It now reads "The lower leg…". These high-density sampling legs are introduced as occurring at mid-cloud levels. This section is only discussing these two mid-cloud legs shown in Figure 10.

24.   Section 4.3.1 - Consider references to Leaitch et al. (JGR, 1986; https://doi.org/10.1029/JD091iD11p11821) and Hill et al. (JGR, 2007; doi:10.1029/2006JD008002) that discuss measured enhancements in nitrate in cloudwater collected in convective cloud.

Thank you. These are useful references for nitrate sources and have been included.

25.   Line 729 – The Quinn reference appears to be missing.

Thank you for identifying this. It has been added.

26.   Lines 734-747 – The aim of this paragraph seems to be a reference to justify the lab experiment some of the authors are conducting. Otherwise, it tells us nothing because there are so many uncertainties associated with the NH4+ measurements and so many related processes to consider. This paragraph complicates the paper and I suggest removing it.

In many other instances the reviewer suggests we add additional discussion and analysis even though to do so may extend beyond the available observations, yet on this aspect the recommendation is to remove it entirely. Yes, we agree it is complicated because there are many processes that affect both instrument closure as well as the skill with which we might expect to be able to explain the interplay between the aerosol and cloud data. But to ignore it just because it is complicated does not seem like a good justification.

The section discussing ammonium is quite short in comparison with other sections in recognition of the fact that there is a limit to what we can solidly claim based on instrument questions that need to be resolved. A reader (or reviewer) may legitimately ask the question as to why we have left out any discussion about the ammonium measurements.

In light of the fact that Reviewer 1 recommended that we make this paper a Measurement Report, we are even more strongly in favor of reporting these measurements.

27.   Line 806 – I think it is reasonably well known that oxalic acid can be unstable, depending on temperature, but your statement that oxalate is not thermodynamically

favored on sea salt particles needs a reference.  Are you referring to oxalate after reaction of oxalic acid with NaCl?

This comment is a little unclear so if we have misunderstood the reviewer's intent we will happily reconsider.  Implicit in the discussion of aerosol particle solutes is that these ions exist in an aqueous solution corresponding to conditions found in the tropical marine boundary layer.  Organic-rich particles may involve more complex phase partitioning, but for sea salt and ammonium bisulfate (or sulfuric acid) particles it seems reasonable to assume that an aqueous phase exists. The partitioning is then a function of the aerosol droplet pH and Henry's Law. We have already provided a reference for oxalate/oxalic acid partitioning when ammonia availability is low (Paciga et al., 2014).  In light of this comment, we have made a slight modification to reinforce the fact that particles sampled by the PILS are acidic in Cases I and II: "…oxalate was not detected on **the acidic** particles sampled by the PILS in Cases I and II…".  There is no implication that oxalic acid reacts specifically with NaCl, rather that it seeks equilibrium with the ions in aerosol droplets.  The addition of sulfate to small SS particles is likely a requirement to acidify these particles, but it is reasonable to assume that sea salt and sulfate do not remain externally mixed upon aging (e.g., coalescence scavenging of sulfate-containing droplets during interactions in cloud, condensation of sulfuric acid vapors on sea salt, aqueous production).  We cannot explicitly state that there is enrichment of small sea salt particles with (nss) sulfate because we do not have single particle composition measurements, but this is what is inferred based on the measurements that we do have.

28.   Line 818 – Does this consider the AMS-CE and CVI collection efficiencies as well as size constraints on CW sampling?

The quick answer is no.

First, there are no CVI data presented here at all and this is stated in L174-176.  Regarding AMS-CE: for this to manifest in this comparison, there would have to be a CE difference between OA and sulfate for particles that have some degree of internal mixing.  For Cases I and II, the CE for sulfate (using comparison with PILS) was found to be ~1, which is in line with conventional assumptions about the CE for OA.  Case III is perhaps more challenging, since the sulfate CE was found to be 0.7-0.74 and so this may result in a differential CE affecting the AMS $f_cOA/SO_4$ shown in Fig 13b.  However, this has already been stated in L830.  Finally, the droplet size sensitivity for the CW samples is an important factor but not something that we could quantify in this comparison in any meaningful way.  The assumption is that the CW samples reflect the composition of the bulk LWC and reject any contribution from interstitial particles.  Roll-off in the efficiency of collection of small cloud droplets could create a bias where this compositional subset is under-represented.  In Section 2.3, we do report the reduced collection efficiency for D<20 µm from Crosbie et al., 2018 and note the large drops are efficiently collected (which was previously not characterized).  Case III is likely to be more susceptible to this potential bias because the drop sizes were smaller, and so this has been added to the discussion:

"A second explanation for the apparent exceedance of OA is that the scavenged fraction may be lower for the less hygroscopic organic matter compared to sulfate, **or that organic-**

**rich droplets may be smaller causing them to be under sampled in CW (because of decreased collection efficiency at D < 20 µm)."**

29. Lines 831-832 – This is a good example of something that could stand to have more focus and detail. There are many indications that OA can be distributed more broadly across the size spectrum than SO4=.

It is not clear which indications the reviewer is referring to. OA is the main component of the submicron mass and so its mass size distribution should approximate that of the LAS volume size distribution.

We have revised this section to add more detail but unfortunately the mechanisms we think may contribute do not fully explain the observed behavior. In light of calculations performed to support high fraction of mass activated and the large volume mode, we have refined statements made here about the potential significance of size dependent composition because it seems that, using any reasonable assumptions about mixing state, size-dependent composition, and the expected peak supersaturations, we cannot reconcile the lower organic content of the cloud from calculations at activation.

30. Lines 838-841 – Interesting, but the analysis is again insufficient.

This comment is challenging to action. These lines are part of the same discussion that was critiqued in Comment 29. We have tried to expand the text in relation to both comments but there is a limit to the utility of additional detail, since it is starting to become overly speculative.

We think it is important to report the differences in the organic:sulfate ratio for clouds and aerosol between the smoke case and the other two, but we can really only list the factors (instrument/measurement, physical, chemical) that may contribute to the observed difference. We have added some more detail including references to studies that have analyzed similar microphysical mechanisms pertaining to cyclical trajectories in clouds and the effects on spectral broadening and deactivation.

We are interested in running parcel simulations to investigate our claims about sulfate/organic and sulfate/seasalt differential scavenging under the action of entrainment in cumulus clouds, as well as the behavior of cloud water composition in downdrafts. To our knowledge, past work on this topic has been solely focused on cloud droplet microphysics and not on mass budgets of cloud water solutes. However, we think this is out of scope for the current manuscript, especially under its classification as a "Measurement Report".

31. Lines 904-905 – about 25% of the total carbon was measured in the CW samples.

No that is not correct. But we can now see that the statement was ambiguous and have revised it. It now reads:

"The carbon from measured CW speciated organic ions comprised 25% of TOC, a result that was remarkably consistent between the three cases, despite differences in the relative abundance of the measured organic ions."